# Global change in brain state during spontaneous and forced walk in *Drosophila* is composed of combined activity patterns of different neuron classes

**Sophie Aimon[1]\*, Karen Y Cheng[1,2], Julijana Gjorgjieva[1,3], Ilona C Grunwald Kadow[1,2]\***

[1]School of Life Sciences, Technical University of Munich, Freising, Germany; [2]University of Bonn, Medical Faculty (UKB), Institute of Physiology II, Bonn, Germany; [3]Max Planck Institute for Brain Research, Computation in Neural Circuits, Frankfurt, Germany

**\*For correspondence:** aimon.sophie@gmail.com (SA); ilona.grunwald@ukbonn.de (ICGK)

**Abstract** Movement-correlated brain activity has been found across species and brain regions. Here, we used fast whole brain lightfield imaging in adult *Drosophila* to investigate the relationship between walk and brain-wide neuronal activity. We observed a global change in activity that tightly correlated with spontaneous bouts of walk. While imaging specific sets of excitatory, inhibitory, and neuromodulatory neurons highlighted their joint contribution, spatial heterogeneity in walk- and turning-induced activity allowed parsing unique responses from subregions and sometimes individual candidate neurons. For example, previously uncharacterized serotonergic neurons were inhibited during walk. While activity onset in some areas preceded walk onset exclusively in spontaneously walking animals, spontaneous and forced walk elicited similar activity in most brain regions. These data suggest a major contribution of walk and walk-related sensory or proprioceptive information to global activity of all major neuronal classes.

## Editor's evaluation

This paper expands on prior work by using whole-brain calcium imaging in *Drosophila* to examine how spontaneous and forced walking and turning affect neural activity in the brain. The measurements presented will serve as a valuable resource for the fly systems neuroscience community and suggest many testable hypotheses that may serve as the basis for future studies. Analyses of the data are solid, and presented with appropriate caveats. This article will be of interest to neuroscientists engaged with the central problem of how behavior modulates neural activity.

## Introduction

Growing evidence from nematodes to mammals shows that ongoing behavior affects brain activity globally (*Kaplan and Zimmer, 2020*; *Parker et al., 2020*). Using a combination of imaging and neuropixel recordings in awake, behaving mice, recent work showed that multiple dimensions of (spontaneous) behavior, including facial or body movements, are represented brain-wide, allowing the integration of external or internal stimuli with the current behavioral state (*Macé et al., 2018*; *Musall et al., 2019*; *Stringer et al., 2019*). Brain-wide imaging at single cell resolution of calcium activity

in *C. elegans* and larval zebrafish indicated that as in mammals, multiple aspects of behavior and motor activity are represented across the brain including areas thought to be dedicated to primary sensory information processing (*Kato et al., 2015*; *Marques et al., 2020*; *Vladimirov et al., 2014*). Importantly, such brain-wide representations of motor states are observed independently of visual or olfactory inputs.

Previous studies suggested a similar phenomenon in insects. For example, in *Drosophila melanogaster*, active flight modulates visual motion processing (*Maimon et al., 2010*). Similarly, visual horizontal system cells encode quantitative information about the fly's walking behavior independently of visual input (*Fujiwara et al., 2017*). Beyond primary sensory brain areas, several types of dopaminergic neurons innervating the fly's higher brain center, the mushroom body (MB), show activity highly correlated with bouts of walking (*Siju et al., 2020*; *Zolin et al., 2021*). Importantly, whole brain imaging revealed that behavior-related activity occurred in most brain regions and was independent of visual or olfactory input (*Aimon et al., 2019*; *Mann et al., 2021*; *Schaffer et al., 2021*).

These and other studies provide convincing evidence for brain-wide signatures of behavior across species. However, the identity of the specific neuron types and neurotransmitters involved as well as where and how behavior-related information is relayed to the brain remain largely unanswered questions. Complementary to other animal models, the fly provides opportunities to study adult, global brain states associated with behavior at high temporal and spatial resolution due to its small brain size (*Aimon and Grunwald Kadow, 2019*). In addition, recent electron microscopy (EM) connectomics highlighted that neural networks spread across the entire brain with many pathways unrevealed by traditional single neuron manipulation experiments (*Scheffer et al., 2020*). Adult *Drosophila* behavior has been studied in detail for many years (*Berman et al., 2016*; *Calhoun et al., 2019*; *DeAngelis et al., 2019*; *Geurten et al., 2014*; *Katsov et al., 2017*; *Mendes et al., 2014*; *Mueller et al., 2019*; *Tao et al., 2019*; *Tsai and Chou, 2019*), but the resulting or underlying brain-wide activity for even just roughly defined behavioral states such as resting or walking remains mostly unknown.

Movement in adult *Drosophila* is thought to be controlled by descending neurons connecting the brain to the ventral nerve cord (VNC), which contains local pattern generators responsible for movement (*Bidaye et al., 2020*; *Bidaye et al., 2014*; *Emanuel et al., 2020*; *von Philipsborn et al., 2011*; *Zacarias et al., 2018*). This top-down view is challenged by other studies suggesting that behavioral control is decentralized with feedback loops involving the brain (*Schilling and Cruse, 2020*; *Sims et al., 2019*), or even that many behaviors could be locally controlled by neurons in the VNC without involving the brain (e.g., decapitated grooming *Hirsh, 1997*).

Here, we build on our previous work (*Aimon et al., 2019*) using fast whole brain imaging during ongoing behavior to more comprehensively unravel the spatial and temporal relationship among movement, brain state, and neural network activity across multiple brain structures and neuronal subtypes. We first look at which regions and neurons expressing different neuromodulators are correlated to walk and turn. We next characterize the timing of activation at behavior transitions between rest and walk and find that walk-related activity in most brain regions starts at or after the transition. Nevertheless, several activity components, for example, in the posterior slope, show increased activity before walk onset. This is consistent with the hypothesis that brain activity in most regions originates primarily from efferent or proprioceptive feedback, and that few brain regions are directly involved in top-down movement control. To test this hypothesis, we compare global brain activity during forced and spontaneous walk and find them to be similar in most brain regions with select areas preceding exclusively spontaneous walk. Our results suggest that walk elicits a global change in brain state composed of the activity of all brain regions and major neuron types.

## Results

### Whole brain imaging reveals broad activation during walk

To image whole brain activity during spontaneous behavior, we fixed the fly's head to a holder, opened the posterior head capsule for optical access while leaving the legs free to move (*Woller et al., 2021*; *Figure 1A*), and used a ball as a walkable surface (*Seelig et al., 2010*; *Figure 1A*). We expressed GCaMP pan-neuronally and imaged calcium transients in the neuropil as a proxy of neuronal activity in the whole brain using fast light field microscopy (LFM) as described previously (*Aimon et al., 2019*). Briefly, a multilens array captures an image of the entire brain. These raw images

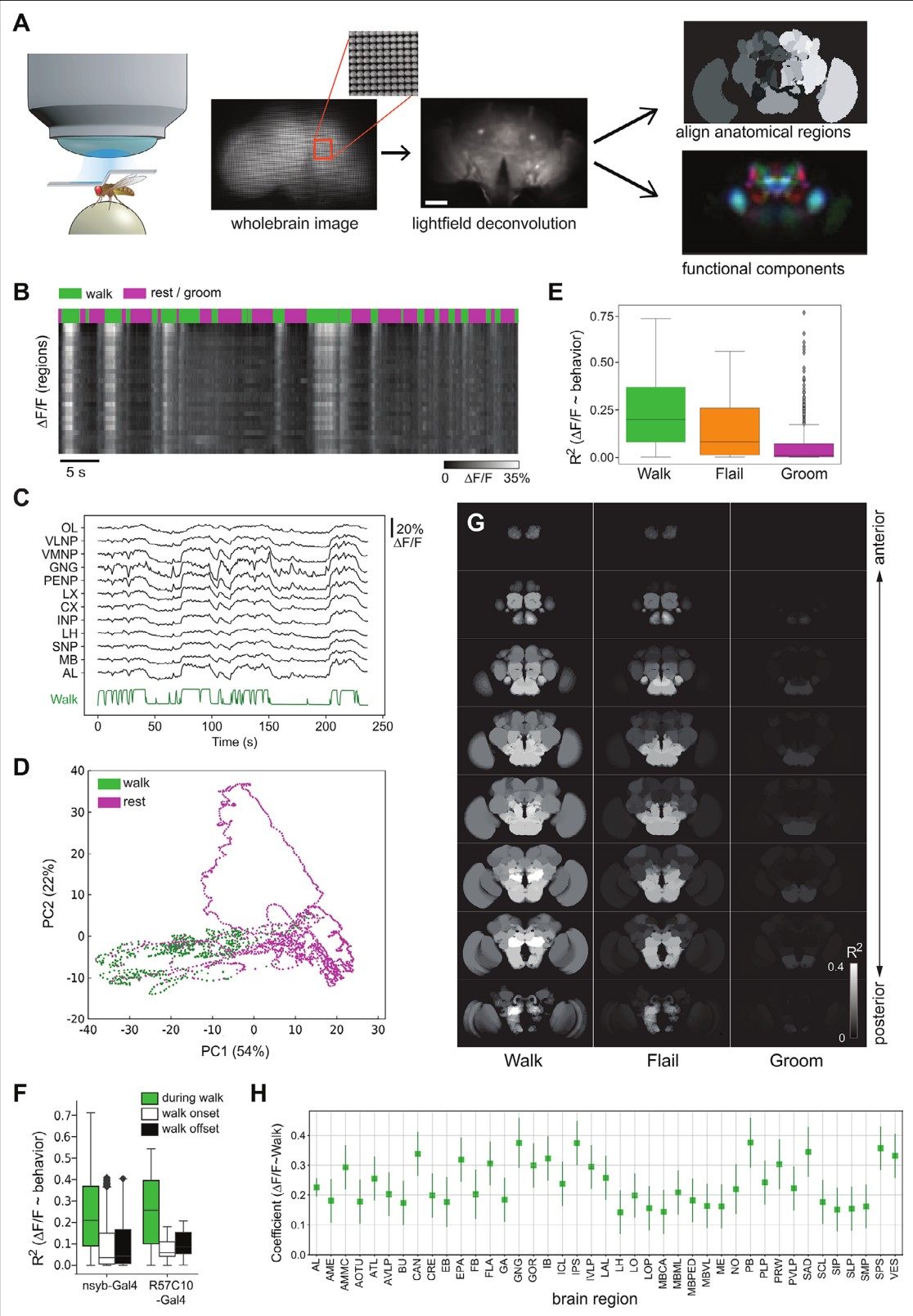

**Figure 1.** Global brain activation during walk. (**A**) Schematic overview of the preparation and analysis method. Please see methods for details. (**B**) Raster plot of the activity of regions. Top panels depict walking bouts in green and rest or grooming in magenta. Lower panel shows calcium activity elicited throughout the experiment. The brighter the higher the calcium transients. Mean forward speed: 5.6 mm/s, mean angular velocity: 0.4 rad/s. Bar is 60 μm. (**C**) Sample traces (Δ*F/F*) for different brain regions relative to walk (green). (**D**) First two principal components from whole brain activity color coded

*Figure 1 continued on next page*

*Figure 1 continued*

with behavior (see additional examples in *Figure 1—figure supplement 1*). (**E**) $R^2$ for regression of single regions with single behaviors (all regions were pooled, but p-values are obtained after averaging regions for each fly. Walk: $N = 16$, Flail: $N = 7$, Groom: $N = 6$). Mann–Whitney $U$-test Bonferroni adjusted p-values: Walk vs. Flail: 0.085, Walk vs. Groom: 0.011, Flail vs. Groom: 0.26. Center line, median; box limits, upper and lower quartiles; whiskers, ×1.5 interquartile range; points, outliers. (**F**) $R^2$ of single region activity regression with walk, walk onset and walk offset (all regions were pooled, nsyb-Gal4: $N = 16$, GMR57C10-Gal4: $N = 4$). Regressors for walk onset or offset are Dirac functions convolved with the GCaMP response (see Methods). Box plots show: center line, median; box limits, upper and lower quartiles; whiskers, ×1.5 interquartile range; points, outliers. Mann–Whitney $U$-test for the two genotypes grouped (comparison of fly-wise averages): walk vs. walk onset: $p = 3 \times 10^{-5}$, walk vs. walk offset: $p = 6 \times 10^{-5}$. (**G**) Z-stack map of $R^2$ median (Walk: $N$ (flies) = 16, Flail: $N = 7$, Groom: $N = 6$) for regression between single region activity and walk, flail or groom (see values in *Figure 1—figure supplement 1D-F*). (**H**) Coefficient of single region' activity regression with walk. All regions' 95% CI ( bars) are above zero and all adjusted p-values <0.001 (Benjamini–Hochberg correction). $N = 16$.

The online version of this article includes the following figure supplement(s) for figure 1:

**Figure supplement 1.** Correlation of regional activity with walk.

were subsequently processed to reconstruct the brain volume, corrected for movement artifacts, and aligned to a template to identify the brain regions or activity components with behavior-correlated calcium changes (*Figure 1A*). This resulted in recordings of neuropil at a typical scale of 5 µm (*Aimon et al., 2019*). To characterize the relationship between neuropil activity and behavior, we used linear models ($\frac{\Delta F}{F} = f(behavior)$), generating a coefficient and $R^2$, which have complementary properties. The $R^2$, representing the fraction of variance explained by the behavior, will decrease with noisier signals (such as those from flies with low GCaMP expression levels). By contrast, the coefficient, representing the intensity and sign of the relationship between activity and behavior, will not have the same dependance to noise (its value will be less precise with higher noise). While LFM allows for very fast imaging frame rates (up to 200 Hz for the whole brain), signal-to-noise ratio decreases with imaging speed. We used a variety of GCaMP sensors (*UAS-GCaMP6s/6m/6f/7s/7f*) and frame rates to capture both high speed and low signal-to-noise transients. With our data ($N = 12$ flies), we found no significant difference between the $R^2$ at different frame rates (5–98 Hz) for flies expressing GCaMP pan-neuronally (*Figure 1—figure supplement 1A*), and thus pooled these data for the analysis below regarding global brain state.

We observed a strong increase in neuronal activity across the brain during bouts of walking compared to during resting or grooming (*Video 1*, *Figure 1B, C*). Accordingly, in experiments where the fly was both walking and resting, the first principal component (PC1) of the whole brain activity data was strongly correlated to walk (*Figure 1D*). By aligning whole brain imaging data to the JFR2018 fly brain template using landmark registration, and by averaging activity in each large anatomically defined region (see Methods), we also found that the global activity of individual brain regions correlated with walk (*Figure 1B, C*). Neuronal activation followed movement bouts at high temporal precision (limited by the temporal resolution of the calcium reporter) inconsistent with activity generated by slower general arousal such as wake and sleep (*Figure 1B, C*). However, calcium responses were still a little slower than some of the fast changes in walking, even after convolution of walking with the GCaMP impulse response, which suggests that faster probes could capture finer dynamics in further studies (*Figure 1C*). Nevertheless, using walking behavior we were able to explain ~20% of all variances observed in the experiments ($R^2$

**Video 1.** Movie of pan-neuronal activation during walk and groom (accelerated).
https://elifesciences.org/articles/85202/figures#video1

**Table 1.** Summary statistics of $R^2$ dependence on different factors where $R^2$ is obtained from regressing a single regionally averaged brain activity with a single behavior.

| | sum_sq | PR(>$F$) |
|---|---|---|
| GAL4 | 0.002197 | 7.58E−01 |
| Behavior | 12.215185 | 1.64E−106 |
| RegionNames | 10.372902 | 4.13E−65 |
| UAS | 2.083138 | 2.13E−18 |
| Behavior:RegionNames | 1.45213 | 9.17E−01 |
| Residual | 67.904784 | NaN |

median = 0.194, *Figure 1E*). Moreover, the global activity correlated primarily with walk and not with the start or end of walk (*Figure 1F*).

We next mapped the differences in activity during different behaviors to smaller brain regions to understand their spatial organization. Strikingly, unlike grooming, which resulted in local activation of ventral brain areas consistent with (*Hampel et al., 2015* and *Robie et al., 2017*), all brain regions were significantly positively correlated with walking (*Figure 1E, G*). To compare walk- vs. flailing-related neural activity, we also carried out experiments without a walking substrate, so that the fly moves its legs freely in the air. While we still observed an increase in activity during bouts of flailing, the brain was not as consistently globally activated as during walking in our hands (*Figure 1E, G*, see also *Schaffer et al., 2021*). We used one linear model to characterize the correlation of brain activity with walking ($\frac{\Delta F}{F} = f\left(walk\right)$), generating a coefficient and ($R^2$) and an additional linear model to determine how this correlation depended on the brain region ($R^2 = f\left(region\right)$ and $Coefficient = f\left(region\right)$). To ensure our results were not biased by heterogeneous expression in the pan-neuronal transgenic driver line, we used two different pan-neuronal lines (*nsyb-Gal4* and *GMR57C10-Gal4*). The linear model (*Table 1*) showed that there was no significant effect of the Gal4-driver. We found a small but significant effect of the GCaMP version used, likely reflecting the higher signal to noise ratio for GCaMP6s and GCaMP7s, and thus kept using the model to take this effect into account for the rest of the study. Using this model, we quantified the effect of the distinct brain regions on both the normalized coefficient and the $R^2$ of the regression of regionally averaged activity with the behavior regressor (*Figure 1H*, *Figure 1—figure supplement 1 D-J*). While all brain areas were significantly activated during walk (*Figure 1G, H*, *Figure 1—figure supplement 1C*), only the ventral regions (AMMC, GNG, IPS, IVLP, and SAD; see *Table 2* and *Ito et al., 2014* for meaning of acronyms) showed significant correlation with grooming (*Figure 1—figure supplement 1F,H*). Flailing represents an intermediate state with many brain regions activated but less consistently as compared to walk (*Figure 1G*, *Figure 1—figure supplement 1E,G* ).

To test whether part of the observed global brain activity could be due to visual input coupled to behavior, for example, by the fly seeing the optic flow from the ball (*Borst et al., 2020*; *Suver et al., 2016*), or unexpectedly fixed reflections from the environment (*Creamer et al., 2018*), and to use an acute approach complementary to developmentally blind norpA mutant flies (*Aimon et al., 2019*), we performed the same experiments but covered the fly's eyes with black nail polish to prevent outside light from activating the fly's photoreceptors. With this strongly limited visual input, we still observed a similar, global activity pattern indicating that visual input is at best a minor contributor to our observed wide brain activity (*Figure 1—figure supplement 1B*). This result also suggests that the global increase in activity during walk is not due to a mismatch between an actual and a predicted visual stimulus.

Given that air-supported balls can show erratic movements due to air turbulences that could cue the fly to change its behavior (*DeAngelis et al., 2019*; *Sen et al., 2019*), or impose constraints on the animal's posture, we performed parallel experiments usin an unsupported styrofoam ball held by the fly (see Methods). The comparison of the two datasets revealed a significant difference in $R^2$ for global brain activity (*Figure 1—figure supplement 1I*), possibly reflecting differences in surface material, or erratic movements of the air-supported ball unrelated to the fly's movements confounding the walk regressor. While the $R^2$ was globally different, distribution across brain regions was similar between the two substrate types (cosine similarity = 0.98, *Figure 1—figure supplement 1J*). We thus combined the datasets for most analysis below.

Together these data confirm that walking behavior induces a change in global brain activity (*Aimon et al., 2019*) with most brain regions showing highly temporally correlated activity during bouts of spontaneous walk.

## Inhibitory and excitatory neurons, as well as aminergic neurons are recruited during walk

We next asked what major neuron types underlie this global change. We expressed GCaMP6 exclusively in GABAergic inhibitory neurons (*GAD1-Gal4; UAS-GCaMP6m*), glutamatergic neurons (*Vglut-Gal4;UAS-GCaMP6m*), and excitatory cholinergic neurons (*Cha-Gal4;UAS-GCaMP6m or f*). Using the same approach and analysis as described above, we detected an increase in global brain activity for excitatory and inhibitory types of neurons (*Figure 2A,B*), closely following individual walk bouts

 

**Table 2.** List of components with potential underlying candidate neurons. Most matches are speculative except those in bold that correspond to high confidence matches due to the lack of other Gal4-positive neurons in the region.

| Short name | Description | Examples of matching (Flycircuit) neurons | Present in Gal4 line (detected in ≥5 flies) |
|---|---|---|---|
| **Antennal lobe and mushroom body** | | | |
| AL | Whole antennal lobe | Gad1-F-000394; Gad1-F-100601; Gad1-F-800129; VGlut-F-900126 | Nsyb, GMR57C10, Vglut, Gad |
| PN | Antennal lobe projection neuron | VGlut-F-500486 | Nsyb, GMR57C10, Cha, Vglut, Gad |
| MultiGl | Whole antennal lobe and PN-like projections | Trh-F-600017 | TH, Trh, TDC |
| PNv | Antennal lobe projection neuron, ventral tract of the lateral horn | VFB_00101133 | Nsyb, GMR57C10, Cha, Vglut, Gad |
| PN-KC | Antennal lobe projection neuron and kenyon cell in the same component | | Nsyb, GMR57C10, Cha, Vglut |
| KCab | Alpha-Beta Kenyon cell | Cha-F-300226; Cha-F-100049; fru-F-000026; Vglut-F-100284; Gad1-F-100014; | Nsyb, GMR57C10 |
| KCapbp | Alpha'-Beta'; Kenyon cell | Gad1-F-100024; Trh-F-200069; | Nsyb, GMR57C10 |
| KCg | Gamma kenyon cells | fru-F-000006; Vglut-F-100359; Gad1-F-100021; | Nsyb, GMR57C10 |
| Beta1Betap1 | Beta1 and/or Beta'1 mushroom body compartment | PAM10(B1)_L (FlyEM-HB:1328522741) [VFB_jrchk385] | TH |
| Beta2Betap2 | Beta2 and/or Beta'2 mushroom body compartment | PAM02(B'2a)_L (FlyEM-HB:1295566429) | Nsyb, Gad, TH |
| **Gamma1** | **Gamma1 mushroom body compartment** | PPL1-gamma1-pedc | TH |
| **Gamma2** | **Gamma2 mushroom body compartment** | PPL1-gamma2-alpha'1 | TH |
| **Gamma3** | **Gamma3 mushroom body compartment** | MBON-γ3, PAM-γ3 | Gad, TH |
| **Gamma4** | **Gamma4 mushroom body compartment** | PAM-γ4 | TH |
| **Gamma5** | **Gamma5 mushroom body compartment** | PAM-γ5 | TH |
| SLP-Alpha | Alpha or Alpha' lobe with projection through the superior lateral neuropil | VGlut-F-500002 | Vglut |
| **Alpha1** | **Alpha1 mushroom body compartment** | PAM-alpha1 | TH |
| **Alpha2** | **Alpha2 mushroom body compartment** | PPL1-alpha'2alpha2 | TH |
| **Alpha3** | **Alpha3 mushroom body compartment** | PPL1-alpha3 | TH |
| **Alphap3** | **Alpha'3 mushroom body compartment** | PPL1-alpha'3 | TH |
| **Superior neuropil** | | | |
| SCLtract | Tract linking both superior clamp | Trh-F-200082,Trh-F-100051 | Trh |
| CLvert | Lateral part of the superior clamp | TH-F-000023 | Nsyb, TH |

*Table 2 continued on next page*

 

*Table 2 continued*

| Short name | Description | Examples of matching (Flycircuit) neurons | Present in Gal4 line (detected in ≥5 flies) |
|---|---|---|---|
| CL? | Interrogation point surrounding the pedonculus, | Trh-F-300074,Trh-M-700081, DNp32 | TH, Trh |
| CL-LH | Surrounds the lateral horn from the medial and ventral directions | Trh-F-200047 | Trh, TDC |
| CL | Other shapes at the level of the clamp | Gad1-F-700550,Gad1-F-800055 | Nsyb, GMR57C10, Gad, TH, TDC |
| SMPm | Medial part of the superior medial protocerebrum | TH-F-000021; VGlut-F-700286, Cha-F-300251 | Nsyb, GMR57C10, Cha, Vglut, Gad, TH, TDC |
| PPL-SMP | Ventral lateral part of the superior medial protocerebrum, with tracts coming from a posterior lateral cell cluster | TH-F-000019,TH-F-000018,TH-F-000046 | TH |
| SMPl-SIP | Superior intermediate protocerebrum and lateral part of the superior medial protocerebrum | Cha-F-000221,Cha-F-300154, TH-F-300056 | Nsyb, GMR57C10, Cha, Gad, TH, TDC |
| SIP-SMPd | Superior intermediate protocerebrum and dorsal part of the superior medial protocerebrum | fru-F-800063 | Nsyb |
| SIP-FB | Superior intermediate protocerebrum and dorsal layer of the fan-shaped body | Trh-F-100015 | Nsyb, GMR57C10, TH, Trh |
| FB-SN | Broad innervaion of the superior neuropil, and fan-shaped body | OA-VMP3, OA-VPM4 | TDC |
| SLP-SMPproj | Large SMP neuron projecting to ventral regions | Trh-F-700011,Trh-F-000083 | Nsyb, GMR57C10, TH, Trh |
| SLP-SMP | Superior lateral protocerebrum and superior medial protocerebrum | Trh-F-500176, DNp25 | Trh |
| SLP | Superior lateral protocerebrum only | TH-F-100046 | Nsyb, GMR57C10, TH, TDC |
| LH-SLP | Lateral horn and superior lateral protocerebrum | Gad1-F-900346,Gad1-F-600340 | Nsyb, Gad |
| **Central complex** | | | |
| FBcol | Fan-shaped body columns | Tdc2-F-100009; Tdc2-F-100026; Tdc2-F-300001; Tdc2-F-200011; Tdc2-F-100062; Tdc2-F-100016; Gad1-F-900245; Gad1-F-800329; Gad1-F-500513; Gad1-100157 | Nsyb, GMR57C10, Vglut, Gad, TDC |
| FBlayv | Ventral layer of the fan-shaped body | TH-M-300065 | Nsyb, TH |
| FBlaym | Medial layer of the fan-shaped body | TH-F-200055 | Nsyb, TH |
| FBlayd | Dorsal layer of the fan-shaped body | TH-F-200054; Trh-F-300036; Trh-F-400062 | Nsyb, TH, Trh |
| NO | Nodulus or noduli | Cha-F-100429 | Nsyb |
| PB | Protocerebral bridge only | Vglut-F-800282; Vglut-F-600784; Vglut-F-600229; Gad1-F-600267; Gad1-F-100361; Gad1-F-100016; Gad1-F-100593; Vglut-F-000156; Vglut-F-100064 | Nsyb, Vglut, Gad |
| PBfull | Components with full protocerebral bridge | Cha-F-900016, Cha-F-200148 | Nsyb, GMR57C10, Cha |

*Table 2 continued on next page*

*Table 2 continued*

| Short name | Description | Examples of matching (Flycircuit) neurons | Present in Gal4 line (detected in ≥5 flies) |
|---|---|---|---|
| **PB-DA** | **Protocerebral bridge and two dots (maybe cell bodies) at the top of the trachea** | **TH-F-000048** | **TH** |
| BU-PBI-EB | Bulb, ellipsoid body and lateral part of the protocerebral bridge | Gad1-F-900445 | Nsyb, Gad |
| PB-EB | EB-radial and PB glomeruli | Cha-F-500009 | Nsyb, GMR57C10, Cha, Gad |
| EB | Ellipsoid body rings | Trh-F-300095; Cha-F-800146 | Nsyb, GMR57C10, Cha, Vglut, Trh |
| **EB-DA** | **Ellipsoid body and lateral accessory lobe** | **TH-F-100001** | **TH** |
| AOTU-BU | Anterior optic tubercule and bulb | Gad1-F-200712, VGlut-F-400630 | Nsyb, GMR57C10, Cha, Vglut, Gad |
| **Posterior neuropil** | | | |
| IB | Inferior bridge | | Nsyb, Gad |
| ATL | Antler | Adult antler neuron 031 | TH, Trh |
| M-Omega | Posterior ensemble forming an M dorsally and an omega ventrally | TH-F-300078 | TH |
| SPS | Superior posterior slope | VGlut-F-900089, VGlut-F-800136, Cha-F-800003, Gad1-F-900039 | Nsyb, GMR57C10, Cha, Vglut, Gad |
| IPS-Y | Inverse Y shape in the posterior slope | DNb02? | Nsyb, GMR57C10, Cha, Vglut, Gad |
| LAL-PS | Lateral accessory lobe and posterior slope | DNb01? | Nsyb, GMR57C10, Cha, Vglut, Gad |
| **PPM2-LAL-We L,R** | | **TH-F-000000,TH-F-000015,TH-F-000016** | **TH** |
| **PPM2-VMNP-INP L,R** | | **TH-F-000007,TH-F-300058** | **TH** |
| **Lateral neuropil** | | | |
| WPENb | Antennal mechanosensory and motor center and/or Wedge, in the posterior lateral protocerebrum and posterior connection to opposite side | VGlut-F-200005,WPNb, WPNB3#5 (FAFB:4271367) [VFB_001011lp] | Vglut, Gad |
| AMMC-PLP | Antennal mechanosensory and motor center and/or Wedge and branch in the posterior lateral protocerebrum | VGlut-F-400269 | Nsyb, Cha, Vglut, Gad |
| AMMC-WE | Antennal mechanosensory and motor center and/or Wedge | VGlut-F-000138, VGlut-F-400586 | Nsyb, GMR57C10, Vglut, Gad |
| **WE-DA** | **Wedge with two branches forming a large V** | **TH-F-200127,TH-F-000024** | **TH** |
| AVLPonlyproj | Lowest medial part of the anterior ventral lateral protocerebrum projecting ventrally | Cha-F-700097 | |

*Table 2 continued on next page*

*Table 2 continued*

| Short name | Description | Examples of matching (Flycircuit) neurons | Present in Gal4 line (detected in ≥5 flies) |
|---|---|---|---|
| AVLPprojm | Lowest medial part of the anterior ventral lateral protocerebrum projecting ventrally | Gad1-F-500762, Cha-F-400059; Gad1-F-000013 | Nsyb, GMR57C10, Gad |
| AVLPprojl | Lowest lateral part of the anterior ventral lateral protocerebrum projecting ventrally | Cha-F-800125 | Cha |
| AVLPm | Anterior ventral lateral protocerebrum medial part | Gad1-F-900529, Cha-F-800062,Trh-F-400039,Trh-F-400070; Vglut-F-200405*; Vglut-F-900122; Cha-F-400237; Cha-F-200299; Gad1-F-500279 | Nsyb, GMR57C10, Vglut, Gad, TH, Trh, TDC |
| AVLPd | Anterior ventral lateral protocerebrum dorsal part | Cha-F-000424 | Nsyb |
| AVLPshell | Anterior ventral lateral protocerebrum surface | Trh-F-100082 | TH, Trh |
| AVLPsmear | Anterior ventral lateral protocerebrum anterior part | | Nsyb |
| VLPl | Ventro-lateral protocerebrum most lateral part | Gad1-F-900096, Vglut-F-500616,Cha-F-800087 | Nsyb, GMR57C10, Cha, Vglut, Gad |
| PLP-LH | Posterior lateral protocerebrum to the basis of the lateral horn | Gad1-F-500325 | Nsyb |
| PLP | Posterior lateral protocerebrum | Gad1-F-800092 | Nsyb, Gad |
| **Ventral neuropil** | | | |
| PI | Pars intercerebralis | Trh-F-100040, Trh-M-000056 | Nsyb, GMR57C10, TH, Trh |
| PI-PRW | Pars intercerebralis connected to Prow | VGlut-F-600158 | Nsyb, GMR57C10, Gad |
| PRW | Prow | TH-M-000037 | Nsyb, GMR57C10, Vglut, Gad, TH |
| PRW-SLP | Prow and superior lateral protocerebrum | Gad1-F-600213, Cha-F-200258; fru-F-000133; Gad1-F-600213; Trh-F-100091 | Nsyb, GMR57C10 |
| PENP-CL | Periesophageal neuropils and clamp | mALD3_L (FlyEM-HB:822708945) | Nsyb, GMR57C10, Cha, Gad |
| GNGvw | Gnathal ganglia medial and lateral | Cha-F-400186 | Cha |
| GNGm | Gnathal ganglia medial | Cha-F-300235 | Nsyb, GMR57C10 |
| GNGml | Gnathal ganglia medial–lateral | Cha-F-400159 | Nsyb, GMR57C10, Cha, Gad |
| GNGl | Gnathal ganglia lateral | Cha-F-400146 | Nsyb, Cha |
| GNG-AMMC | Gnathal ganglia and on the opposite side antennal mechanosensory and motor center and posterior lateral protocerebrum | VGlut-F-600685 | Nsyb, Vglut |
| GNGva | Ventral anterior part of the gnathal ganglia | | Nsyb |
| vaCells | Ventral anterior cells | TH-F-100049 | TH, Trh |
| **Optic lobe** | | | |

*Table 2 continued on next page*

*Table 2 continued*

| Short name | Description | Examples of matching (Flycircuit) neurons | Present in Gal4 line (**detected in ≥5 flies**) |
|---|---|---|---|
| OL | Optic lobe; mostly medulla and lobulla | | Nsyb, GMR57C10, Cha, Vglut, Gad |
| LOP | Lobulla plate | Cha-F-600161 | Cha, TDC |
| OL-FB | Optic lobe to central regions including the fan-shaped body | | TH |
| OL-PENP | Optic lobe and periesophageal neuropils | Tdc2-F-200056 | Nsyb, Gad, Trh, TDC |
| OL-PLP | Optic lobe and posterior lateral protocerebrum | Cha-F-000316 | Nsyb, Cha, Gad |
| OL-WE | Optic lobe and wedge | TH-F-300030 | TH |

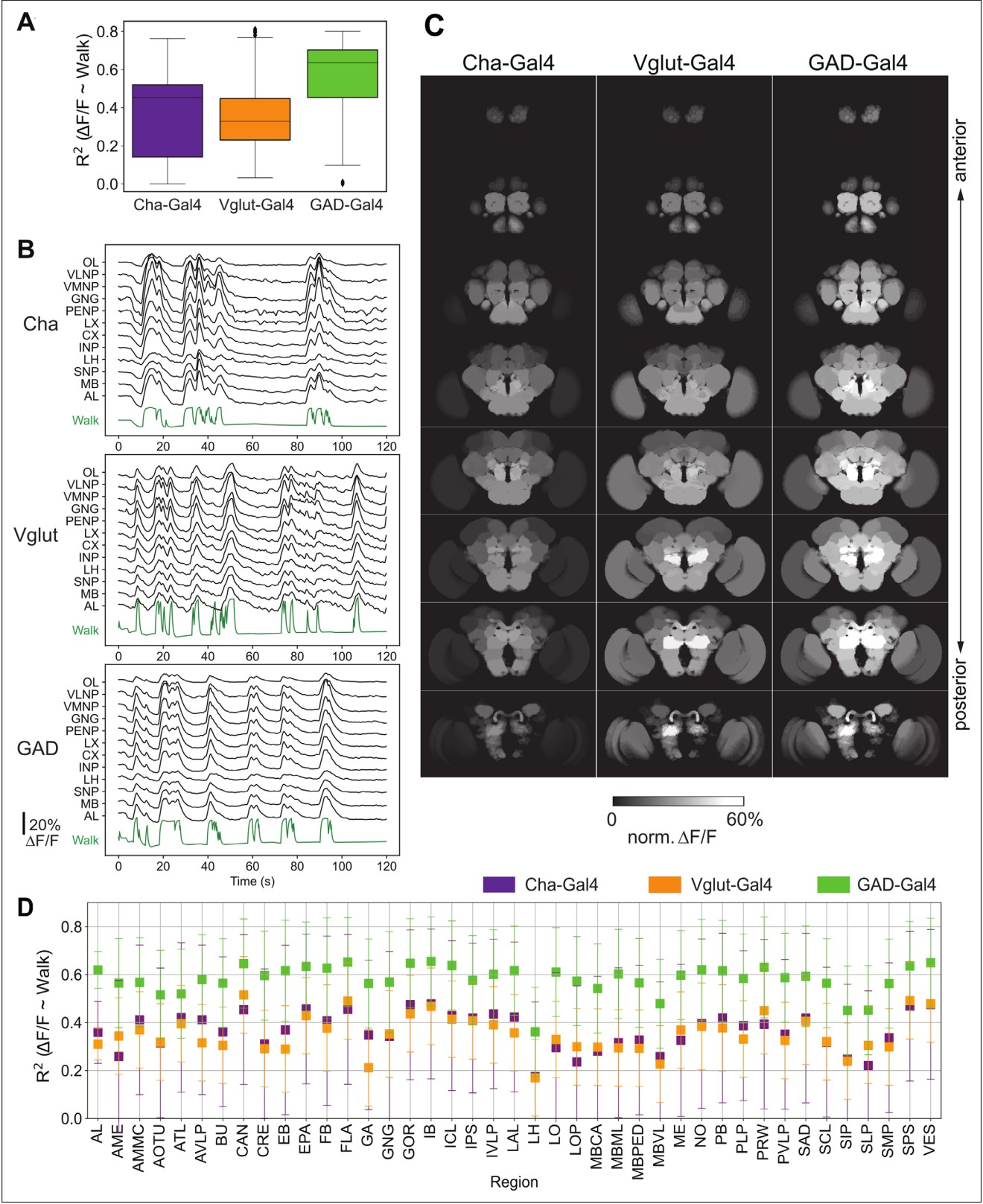

**Figure 2.** Activity of neurons releasing the three major neurotransmitters, glutamate, GABA, and acetylcholine, during walk. *N* = 5 flies for each genotype. (**A**) $R^2$ for regression of single region activity with walk for different genotypes (all regions were pooled). No pairwise comparison of fly-wise averages is significantly different (Mann–Whitney *U*-test). Box plot: center line, median; box limits, upper and lower quartiles; whiskers, ×1.5 interquartile range; points, outliers. (**B**) Maps of activation during walk (regression coefficient of single region activity with walk) for Cha-Gal4, GAD-Gal4, and Vglut-

*Figure 2 continued on next page*

*Figure 2 continued*

Gal4 expressing neurons. See *Figure 2—figure supplement 1* for values and statistical tests. Cosine similarity: Cha vs. Vglut: 0.98, Vglut vs. GAD: 0.98, Cha vs. GAD: 0.99. (**C**) Sample traces ($\Delta F/F$) for different brain regions relative to forward walk (green). See *Figure 2—figure supplement 1* for values. (**D**) $R^2$ for regression of single region activity with walk for Cha-Gal4, GAD-Gal4, and Vglut-Gal4. 95% CI is shown. All values are significantly above zero (Benjamini–Hochberg adjusted *t*-test p-values <0.001).

The online version of this article includes the following figure supplement(s) for figure 2:

**Figure supplement 1.** Correlation coefficients for regression of single region activity with walk.

(*Figure 2B*). As in pan-neuronal data, we mapped neural activity to brain regions using a linear model of regional average activity as a function of walk (*Figure 2C, D*). All regions significantly responded to walk for all genotypes (*Figure 2C, D* , *Figure 2—figure supplement 1*), with high similarity in the distribution across regions (cosine similarity: Cha vs. Vglut: 0.98, Cha vs. Gad: 0.99, Vglut vs. Gad: 0.98) (*Figure 2D*, *Figure 2—figure supplement 1A–C*). Together these data show that both inhibitory neurons and excitatory neurons are activated in most brain regions during walk. Future work will be necessary to determine whether such patterns are driven by all neurons expressing a specific neurotransmitter or a subset of neurons in each region.

We also assessed the role of aminergic neurons in walk, specifically focusing on dopaminergic, octopaminergic, and serotonergic neurons. While we observed an increase in activity for all neuromodulatory neuron types, serotonergic neurons (*Trh-GAL4;UAS-GCaMP6*) were significantly less globally activated than dopaminergic (*TH, DDC* or *GMR58E02-GAL4; UAS-GCaMP*) and octopaminergic (*Tdc-GAL4; UAS-GCaMP6*) neurons during spontaneous bouts of walk (*Figure 3A,D*, *Videos 2–4*). When we mapped activity to brain regions (*Figure 3B*), we found that contrary to the pan-neuronal and broad neurotransmitter line results, maps for individual aminergic lines were distinctly patterned (cosine similarity: TH-DDC vs. Trh: 0.69, Tdc vs. Trh: 0.69, Tdc vs. TH-DDC: 0.93, *Figure 3B, C*). Nevertheless, most regions were significantly correlated with walk for all three aminergic types (*Figure 3F* and *Figure 3—figure supplement 1A,B*). Interestingly, the strongest correlation for serotoninergic neurons was negative and mapped to the anterior ventrolateral protocerebrum (AVLP) region (*Figure 3B* (blue area), *Figure 3E, F*).

## Unsupervised method extracts functional maps matching anatomical structures

We used global brain activity to generate functional maps of the brain and matched their specific shape and location to anatomical maps of subregions and in some cases to single candidate neuron types (*Chiang et al., 2011*; *Ito et al., 2014*; *Table 2*). To this end, we extracted functional components using PCA (principal component analysis) followed by ICA (independent component analysis) to spatially separate PCA maps (see Methods) (*Figure 4A*, *Figure 4—figure supplement 1A, B*). We grouped smaller functional components within a larger brain region (e.g., different antennal and protocerebral bridge glomeruli), if the precision of the alignment of the template did not allow for a clear assignment of individual regions. Interestingly, almost all functional components derived from recorded neuronal activity matches anatomical structures without further subdivisions or blurring of anatomical boundaries. This suggests that the activity within brain regions was more homogenous as compared to the activity in neighboring regions. In one exception, our functional data separated a larger region in the lateral neuropil, the AVLP (anterior ventro-lateral protocerebrum), into smaller subregions (*Figure 4—figure supplement 1A, B*, orange box) suggesting that subregions of the AVLP had different activity signatures. Of note, some components also identified more than one brain region (i.e., PENP-SLP) indicating a strong functional connection between these regions. For example, among the components correlated with spontaneous walk, the PENP-SLP (periesophageal neuropils and superior lateral protocerebrum) and PENP-CL (periesophageal neuropils and clamp) and WPNb-like (bilateral wedge projection neuron-like *Coates et al., 2020*) components connected ventral neuropils (e.g., GNG) to higher areas or vice versa.

As some of the data used to extract maps were from flies expressing GCaMP specifically in inhibitory, excitatory, and neuromodulatory neurons (*Figure 4A*), components could be matched to candidate neurons of the same types (*Table 2*). For dopaminergic neurons, walk-correlated neuronal activity was found, for instance, for components in and around the MB (i.e., γ3 compartment as previously

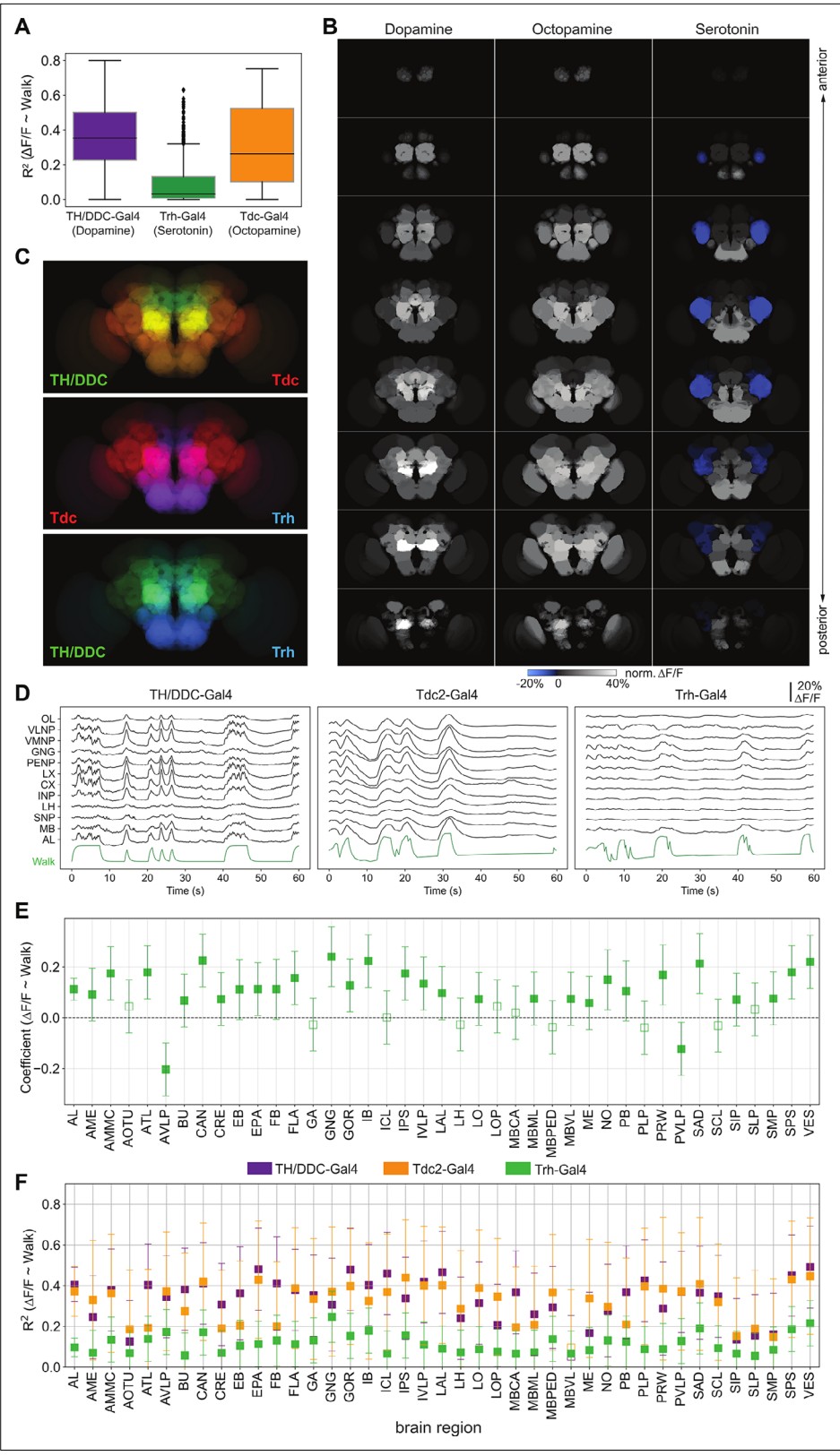

**Figure 3.** Neuromodulatory neurons are strongly and differentially activated during walk. TH/DDC-Gal4: $N = 9$, Tdc2-Gal4: $N = 7$, Trh-Gal4: $N = 6$ flies. (**A**) $R^2$ for regression of single region activity with walk for different genotypes (all regions were pooled). Mann–Whitney $U$-test Bonferroni adjusted p-values: TH vs. Trh: 0.032, TH vs. TDC: ns, Trh vs. TDC: 0.040. Box plot: center line, median; box limits, upper and lower quartiles; whiskers,

*Figure 3 continued on next page*

*Figure 3 continued*

×1.5 interquartile range; points, outliers. (**B**) Maps of activation during walk (regression coefficient of single region activity with walk) for TH-Gal4 and DDC-Gal4 or GMR58E04-Gal4 (dopaminergic neurons), TDC2-Gal4 (octopaminergic neurons), and Trh-Gal4 (serotonergic neurons) expressing neurons. Blue indicates inhibition. (**C**) Overlay of activity maps of two neuromodulators in each panel. Cosine similarity: TH vs. Trh: 0.69, TH vs. TDC: 0.93, TDC vs. Trh: 0.69. (**D**) Sample traces ($\Delta F/F$) for different brain regions relative to forward walk (green). (**E**) Coefficient during walk for different brain regions for Trh-Gal4 expressing serotonergic neurons. All regions are significantly correlated. (**F**) $R^2$ of regression of fluorescence vs. walk for TH/DDC(58E02)-Gal4, Tdc2-Gal4, and Trh-Gal4 expressing neurons. Bars are 95% CI.

The online version of this article includes the following figure supplement(s) for figure 3:

**Figure supplement 1.** Correlation coefficients for regression of single region activity with walk.

observed; *Cohn et al., 2015*; *Siju et al., 2020*; *Zolin et al., 2021*), in the central complex (CX; protocerebral bridge [B-DA] and ellipsoid body [EB-DA-like; *Kong et al., 2010*]), but also in ventral neuropil such as the wedge (WED-DA-like; *Liu et al., 2017*) and neuropil connecting central regions to the optic lobes (*Figure 4A*). For octopamine, one particular component connecting the optic lobe and the periesophageal neuropil (OL-PENP) was strongly correlated with walk, while for serotoninergic neurons the highest $R^2$ was detected for the AVLP and components within (*Figure 4A*). Most components with significant $R^2$ were positively correlated with walk (*Figure 4B, C*), with few exceptions: a specific component within the AVLP mostly detected in the experiment using Trh-Gal4, which we named the AVLPshell due to its shape, was negatively correlated (i.e., inhibited) with walk (*Figure 4B*, *Figure 4—figure supplement 1A–C*).

## Turning modulates activity in specific brain regions and potential neurons

Some functional components activated during walk were mirrored by components in the other half of the brain (*Figure 5A*). We next asked whether these components had differential activity when turning left or right (*Figure 5—figure supplement 1*). To quantify turning, we extracted rotational ball movements (left and right) using its optic flow. The components that were differentially activated during turning were reproducibly found across different flies based on similar position and morphology (*Figure 5A, B*). These included the IPS-Y (inverse Y shape in the posterior slope) and LAL-PS (lateral accessory lobe and posterior slope) as previously mentioned in *Aimon et al., 2019*. For dopaminergic neurons, the components most correlated with turning had a shape closely matching the dopaminergic neuron type known to project to the LAL and WED areas: neurons of the PPM2 cluster, that is, PPM2-LW (PPM2-LAL-WED: PPM2-lateral accessory lobe-wedge; *Mao and Davis, 2009*; *Figure 5A*, right panels; *Video 5*).

## Brain dynamics at transitions between rest and walk

Thus far, we have analyzed neuronal activity during walk, turn, flail, groom, or rest. Next, we analyzed whole brain activity at the transition between rest and walk. Importantly, different GCaMP versions showed similar onset dynamics under our experimental conditions. For improved temporal resolution, we only included datasets recorded at 30 Hz (with a maximum of ±30 ms error between behavior and brain activity) or faster for the entire brain in *Figure 6* (see all individual trials including data recorded at less than 30 Hz in *Figure 6—figure supplement 1*). We normalized activity at the onset of walk

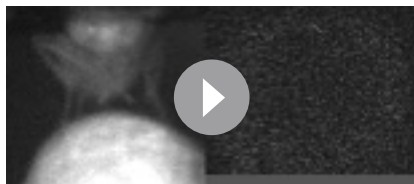

**Video 2.** Movie of TH/DDC-neuronal activation during walk (accelerated).
https://elifesciences.org/articles/85202/figures#video2

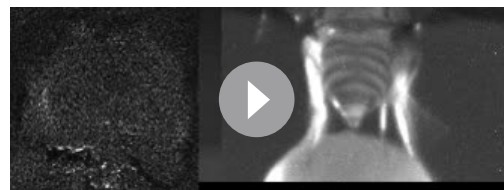

**Video 3.** Movie of TDC2-neuronal activation during walk (accelerated).
https://elifesciences.org/articles/85202/figures#video3

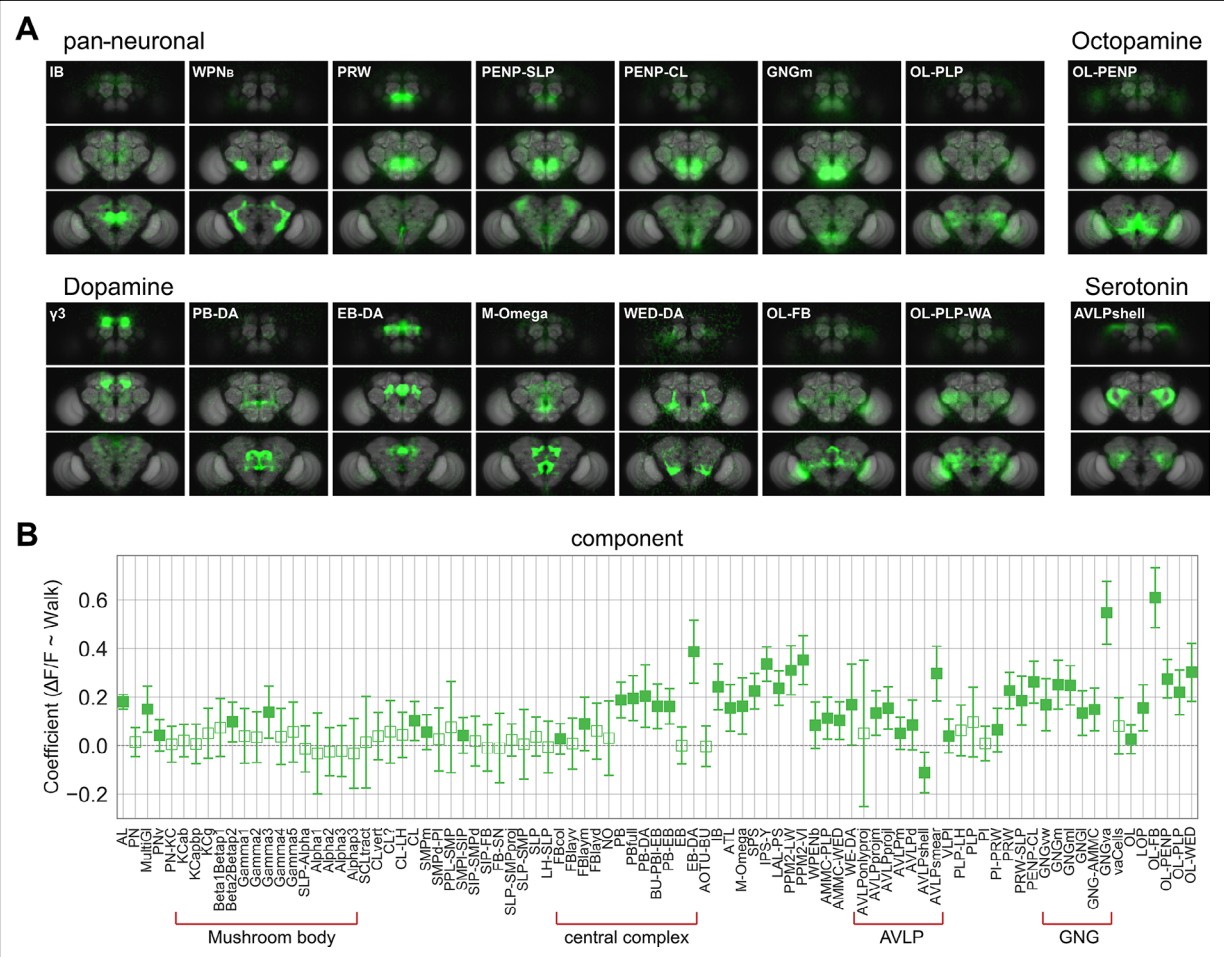

**Figure 4.** Whole brain analysis pinpoints specific subregions responding to walk. (**A**) Images of example components that are significantly correlated with walk (coefficient or $R^2$ 95% CI above zero). Upper left: Components derived from imaging with a pan-neuronal driver. Upper right: Components derived with Tdc2-Gal4. Lower left: Components derived with TH/DDC or GMR58E02-Gal4. Lower right: Components derived with Trh-Gal4. (**B**) Correlation coefficient for component activity vs. walk. $N$ = 58 flies of different genotypes, see table in methods for details. Empty markers correspond to adjusted (Benjamini–Hochberg correction) p-value >0.05 for comparison to 0. Bars are 95% CI.

The online version of this article includes the following figure supplement(s) for figure 4:

**Figure supplement 1.** Component maps and correlation with walk.

and averaged trials. This analysis revealed a more complex picture of neuronal activity with some components being activated before walk onset (*Figure 6*). In particular, we found that regions around the esophagus, such as the posterior slope (IPS-Y, SPS) and the dopaminergic M-Omega component (corresponding to two posterior neuropil regions), as well as the WED-DA-like component showed activity that began to increase significantly before walk onset. In addition, several compartments of the MB and regions in the superior neuropil regions (e.g., SMP and SLP) showed trials with some increase preceding walk (*Figure 6*, *Figure 6—figure supplement 1*). The OL-PLP-WED component was also activated significantly before walk. Most components that mapped to the CX (comprising the protocerebral bridge (PB), fan-shaped body (FB), ellipsoid body (EB), and noduli (NO)) were activated at walk onset but not before (*Figure 6*, *Figure 6—figure supplement 1A*). In particular, the components PB–EB (connecting the PB and the EB) and EB–DA (comprising neurons in the EB and the lateral accessory lobe) became activated once the fly started to walk (*Figure 6*, *Figure 6—figure supplement 1*), consistent with an increase at walk onset of dopaminergic activity in the EB and LAL (*Figure 6—figure supplement 2*). Distinct components mapped to the AVLP displayed very different activities related to walk. While anterior and medial components of AVLP increased in activity at walk onset, the AVLP component that resembled the shape of a shell (AVLPshell) displayed a very clear

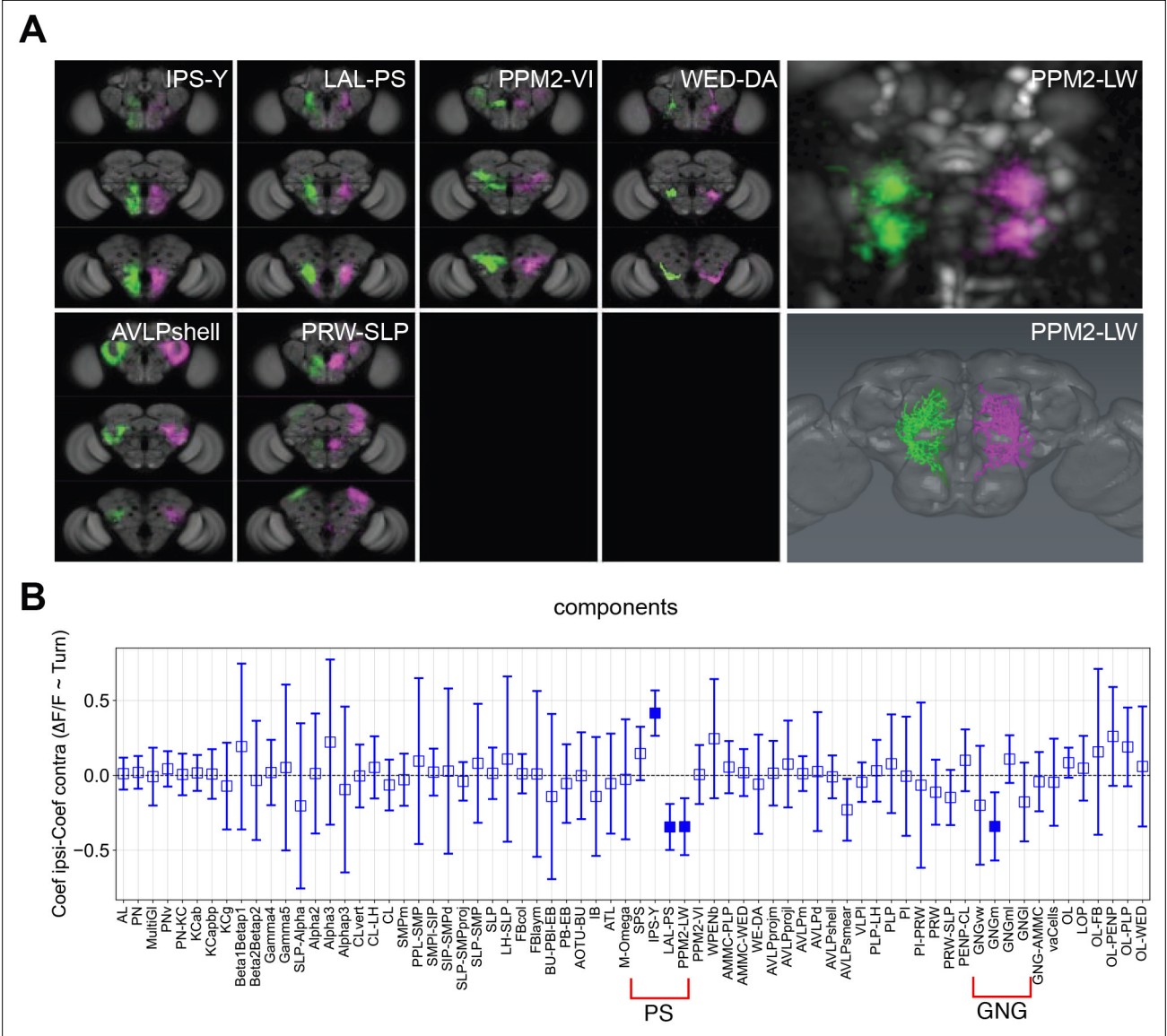

**Figure 5.** Turning activates specific components and candidate neurons. (**A**) Examples of components present in both the left and right hemispheres labeled in different colors (magenta and green). Panels on the right present an example component that could be mapped to a single neuron. Upper right panel: Turning-correlated component, lower right panel: reconstruction of neuron that this functional component was mapped to. (**B**) Difference between the correlation coefficient (normalized Δ*F/F*) for turning on the ipsilateral side and the coefficient on the contralateral side is displayed as a function of the identified components. Positive and negative correlations correspond to components being active more during turn on the ipsilateral side than the contralateral side and the reverse, respectively. See *Table 2* for definition of acronyms. *N* = 58 flies of different genotypes, see table in methods for details. Empty markers correspond to adjusted (Benjamini–Hochberg correction) p-value >0.05 for comparison to 0. Bars are 95% CI.

The online version of this article includes the following figure supplement(s) for figure 5:

**Figure supplement 1.** Turning activates specific components and candidate neurons.

decrease in neuronal activity as soon as the fly started to walk (*Figure 6*, *Figure 6—figure supplement 1*, *Figure 6—figure supplement 2*). Several other components displayed more variable, and at times too variable, dynamics among flies to detect a clear direction (i.e., FBlayv in *Figure 6—figure supplement 1*).

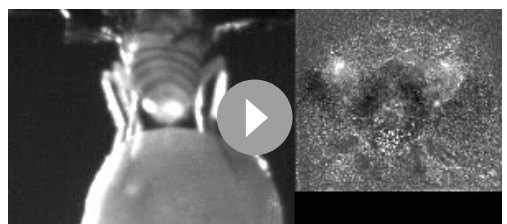

**Video 4.** Movie of Trh-neuronal activation during walk (accelerated).

https://elifesciences.org/articles/85202/figures#video4

## Forced walk recapitulates most activity while forced turning reveals differences

Our data suggest that walking induces a change of activity in most of the brain. But where does this activity come from? We hypothesized that in the extreme case, activity could originate from two opposite sites. First, the activity could arise initially in superior decision-making areas and then spreads across the brain (top-down) or, second, activity is initiated by motor activity and proprioception (bottom-up) and then distributes to higher brain areas. In the latter case, activity would originate in the VNC and move to basal regions of the brain, i.e., the GNG, via ascending neurons (ANs) (*Chen et al., 2023*; *Tsubouchi et al., 2017*). While our component analysis of data recorded at 30 Hz identified several putative 'top-down' components active before walk onset, most activity was more consistent with a 'bottom-up' scenario (*Figures 1 and 6* and *Figure 6—figure supplements 1 and 2*). In some trials, we observed that the activity detected during walk appeared to progress from areas at the base of the brain, that is from GNG to more dorsal areas (*Figure 7—figure supplement 1F*, see *Video 6* for an example trial).

We thus asked whether walk-induced activity could be contributed by axon terminals of ANs. To this end, we expressed a synaptically tethered GCaMP, sytGCaMP6, under the control of a pan-neuronal driver and imaged whole brain activity during walk (*Figure 7A*). Compared to cellular-GCaMP, sytG-CaMP activity was very strong in the GNG, AMMC, and AVLP, regions receiving inputs from the VNC (*Figure 7A*; *Tsubouchi et al., 2017*).

Since such axonal activity, or a fraction of it, could also originate from top-down projections, we performed another experiment to compare neural activity from spontaneous, self-induced walk to that of forced walk. We argued that if global neural activation indeed comes from proprioception of walking or related sensory input from the legs, then we should also observe similar global activation when the fly is forced to walk, without having the possibility to decide to do so as in spontaneous walk. To test this idea, we placed a treadmill controlled by a motor under the fly legs (see *Video 7* and methods). We forced the flies to walk by turning this motor on and off at speeds ranging between 1.5 and 6 mm/s. Similarly to spontaneous walk (*Figure 7B, D*), forced walk induced a change in brain state, with brain-wide activity increases across regions (*Figure 7C, D*, *Video 7*, cosine similarity: 0.993 for $R^2$ and 0.985 for the coefficients), and across components (*Figure 7E*, cosine similarity: 0.87 for $R^2$ and 0.85 for the coefficients). In addition, flies with a surgically severed connection between the brain and VNC walked on the treadmill while showing hardly any activity in the brain (*Figure 7D*, *Video 8*), consistent with the hypothesis that the brain is not necessary for forced walk and walking signals driving whole brain state change originate in the VNC. Importantly, these observations were not only true for pan-neuronal data but also for dopaminergic, octopaminergic, and serotonergic neuronal subsets (*Figure 7—figure supplement 1C–E*). While we did observe some variation between forced and spontaneous walk in the $R^2$ for regression of activity traces with walk, confidence intervals were overlapping for all brain regions (*Figure 7—figure supplement 1C-E*) and cosine similarity was high (TH-DDC: 0.985 for $R^2$, 0.978 for coefficients, Tdc: 0.96 for $R^2$, 0.93 for coefficients, Trh: 0.84 for $R^2$, 0.62 for coefficients) further suggesting that walking, whether spontaneous or forced, elicits a similar global state change of the brain.

By contrast, forced turns elicited different activity signatures from spontaneous turns (*Figure 7F*). We subtracted the $R^2$ for turn of the contralateral side of the brain from the corresponding value of the ipsilateral side and compared these values for individual components. We found that some components switched sign (i.e., IPS-Y) while other components that showed little lateralization during spontaneous turns showed differences for forced turns (i.e.,

**Video 5.** Video sequence showing activation of the PPM2-LW neurons during turning.

https://elifesciences.org/articles/85202/figures#video5

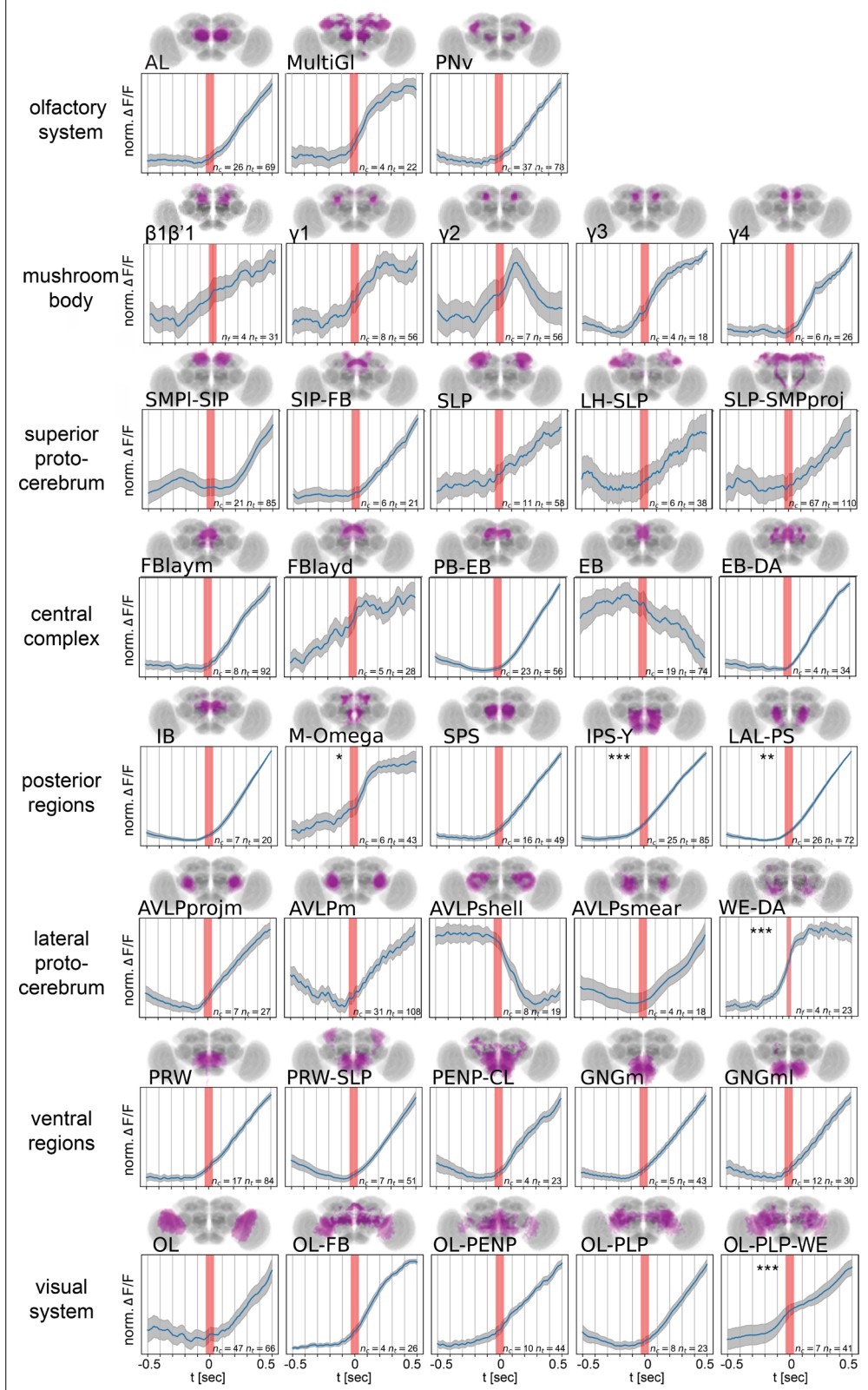

**Figure 6.** Onset dynamics of activity in multiple functional components across the brain at start of walk. Walk-onset triggered average activity for pan-neuronal data in individual active components at 30 Hz or higher temporal resolution. Trials were normalized individually before averaging (see *Figure 6—figure supplement 1* for non-normalized traces and all recording frame rates). Red band indicates walk onset. Note that while most components

*Figure 6 continued on next page*

*Figure 6 continued*

are activated after walk onset, several components show activity before start of walk. Stars correspond to an above zero significance for integrated traces from −0.5 s to start of walk (Wilcoxon one-sided test). Multiple comparison adjusted p-values (Benjamini–Hochberg): *p < 0.05, **p < 0.01, ***p < 0.001. See *Table 2* for definition of acronyms. *N* = 30 flies of different genotypes, see table in methods for details.

The online version of this article includes the following figure supplement(s) for figure 6:

**Figure supplement 1.** Graphs show onset of walk trials for all flies and all transgenic lines at all recording speeds.

**Figure supplement 2.** Normalized trials Δ*F/F* GCaMP fluorescence in brain regions 1 s before to 1 s after walk onset for different anatomically defined brain regions: (**A**) dopaminergic neurons (TH/DDC-Gal4;UAS-GCaMP), (**B**) octopaminergic neurons (Tdc2-Gal4;UAS-GCaMP), and (**C**) serotonergic neurons (Trh-Gal4;UAS-GCaMP).

---

PPM2-VI, GNGm, FBcol) (*Figure 7F*). The PPM2-LW component that showed lateralized activity during spontaneous walk was not lateralized during forced walk.

Given our observation that certain brain regions appear to be activated shortly before walk onset (*Figure 6*), we next compared the activity around walk onset in spontaneous and forced walk more carefully (*Figure 7G*, *Figure 7—figure supplement 1G*). Most components detected in both types of experiments appeared very similar as expected based on the analysis above (*Figure 7E*). Importantly, components with activity prior to spontaneous walk were either activated after walk had been triggered and/or their activity before walk was significantly different between spontaneous and forced walk (e.g., in areas around the esophagus, and in dorsal areas such as the γ3 compartment of the MB and the superior medial neuropil; *Figure 7G*, *Figure 7—figure supplement 1G*).

## Discussion

Work over recent years has revealed that locomotion and movement influence neural activity in many brain areas and many organisms (*Busse et al., 2017*; *Kaplan and Zimmer, 2020*). Importantly, motor activity modulates not only local activity in specific brain regions, but also the global state of the brain. Which neurons and neurotransmitters and neuromodulators underly this activity, how it spreads, and what it means for the animal are still largely unknown (*Kaplan and Zimmer, 2020*). Using fast in vivo whole brain calcium imaging in head-tethered behaving flies, we show that movement elicits a global change in brain activity during spontaneous as well as forced walk. Walk activates several classes of neurons including excitatory, inhibitory, and aminergic, modulatory neurons. Except for serotonergic neurons which are inhibited during walk in some brain areas, we observed neuronal activation across all brain regions at the start of and during walking, but not during grooming or resting. Using PCA/ICA, we mapped neuronal activities to discrete functional components, which we assigned to specific smaller subregions and in some cases even to single candidate neuron types by aligning the activity maps to anatomical data. For instance, we found that maps of dopaminergic activity during turning matched the spatial distribution of specific dopaminergic neurons. Based on the timing of activation and similarity between spontaneous and forced walk-induced brain activity, we propose that locomotion activates the brain by sending movement and proprioceptive information to specific regions (e.g., GNG) from where it activates all brain regions and neuron classes.

### Advantages and limitations of the method

Our approach, using LFM (*Levoy et al., 2006*) to image neuropile activity, has several advantages and limitations, which complement other existing methods. First, the light field speed – significantly higher temporal resolution as compared to sequential scanning methods – allowed us to record whole brain activity at the same time as fast behavior such as walk. Even though GCaMP dynamics and limitations in signal-to-noise ratio rarely permitted us to resolve single action potentials, it helped us detect differences in temporal dynamics and pinpoint brain areas potentially involved in triggering spontaneous walk as

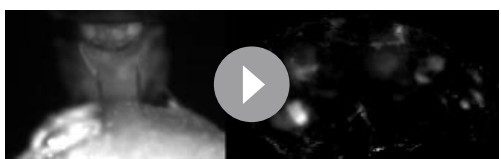

**Video 6.** Video showing neuronal activation moving from ventral to dorsal brain areas. Note that a clear progression of activation was observed only in a fraction of trials.

https://elifesciences.org/articles/85202/figures#video6

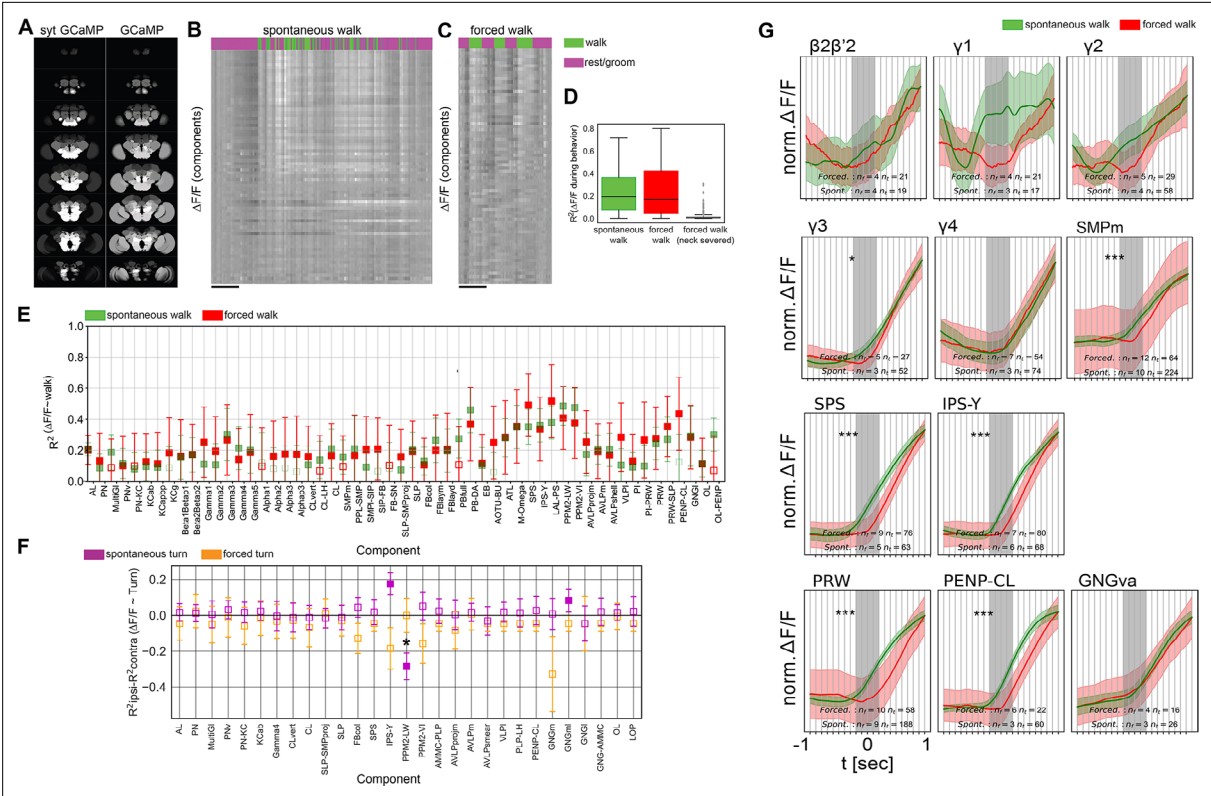

**Figure 7.** Forced and spontaneous walk elicit highly similar global brain activity. (**A**) Z-stacks of map of brain regions activated by walk in flies expressing cytosolic GCaMP (pan-Gal4;UAS-GCaMP6m) or synaptically tagged GCaMP (pan-Gal4;UAS-sytGCaMP6m) ($R^2$ of $\frac{\Delta F}{F} = f\left(walk\right)$). (**B**) Time series of components during spontaneous walk (green) or rest (magenta). Box plots show: center line, median; box limits, upper and lower quartiles; whiskers, ×1.5 interquartile range; points, outliers. (**C**) Time series of active components during forced walk (green) or forced rest (magenta). (**D**) $R^2$ for behavior regression at different conditions (all regions were pooled, spontaneous, forced, forced with severed connection between ventral nerve cord (VNC) and brain). (**E**) $R^2$ for behavior regression for different active components for forced (red) vs. spontaneous (green) walk. Empty markers correspond to adjusted (Benjamini–Hochberg correction) p-value >0.05 for comparison to 0. Bars are 95% CI. (**F**) $R^2$ difference between brain region activity during forced turn on the ipsilateral side and forced turn on the contralateral side (orange). Forced turning speeds ranged from 0.3 to 2 rad/s. Only data with $N \geq 4$ components were analyzed. Lilac shows the difference for spontaneous turns. $N = 58$ flies for spontaneous walk/turn of different genotypes and $N = 26$ flies for forced walk/turn, see table in methods for details. Empty markers correspond to adjusted (Benjamini–Hochberg correction) p-value >0.05 for comparison to 0. Star corresponds to a significant difference between spontaneous and forced turns: Mann–Whitney $U$-test adjusted p-value <0.05 (Benjamini–Hochberg correction). Box plots show: center line, median; box limits, upper and lower quartiles; whiskers, ×1.5 interquartile range; points, outliers. Bars are 95% CI. (**G**) Comparison between activity at walk onset for spontaneous (green) and forced (red) walk for additional components. Individual trials were normalized and averaged for each component. Shaded regions represent the trial SEM (standard error of the mean), nf: number of flies, nt: total number of trials. Gray shaded area indicate walk onset. Stars show the significance of difference in the integral between −0.5 s and the start of walk, and is corrected for multiple comparison (Benjamini–Hochberg): *p < 0.05, ***p < 0.001.

The online version of this article includes the following figure supplement(s) for figure 7:

**Figure supplement 1.** Additional evidence for most activity being induced by walk.

opposed to merely responding to it. Second, the spatial resolution of light field imaging is inferior to that of confocal or volumetric multiphoton imaging. Although these methods also do not allow resolving single neurites with pan-neuronally expressed sensors, light field could in principle make it even more difficult to detect their activity. On the other hand, capturing the whole volume simultaneously makes it easier to unmix signals as we showed with our PCA/ICA approach, which partially compensates for the lower spatial resolution. Importantly, however, averaging the activity in neuropile regions could hide information, and reflect different activity patterns across single neurons. Third, our data are based on observations without genetic or functional manipulation of neurons or circuits. Excitation and inhibition of single neurons or neuron groups are performed frequently in *Drosophila* thanks to its unique genetic tools. We believe that our data complements previous and future functional studies as imaging or manipulation of individual neurons provides only limited insights into

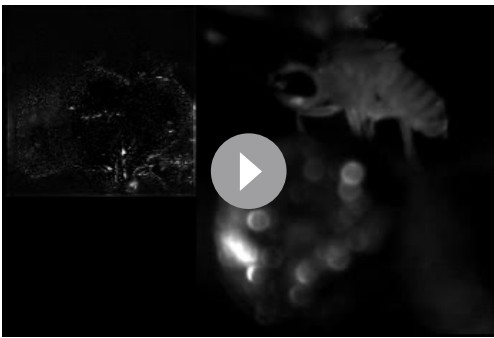

**Video 7.** Video showing whole brain activity of a fly being forced to walk on a rotating rod.
https://elifesciences.org/articles/85202/figures#video7

the role and effect of a neuron in the complex and brain-wide dynamic neural network in which they are embedded. In the future, a combination of single neuron manipulation and whole brain imaging will likely lead to unexpected insights into the relationship between a neuron, global brain state, and a specific behavior. Fourth, we have observed subtler differences between flies that are obvious from individual experiments but difficult to capture quantitatively across a population of animals. These differences might be resolvable by greatly increasing the number of experiments. Given the technical difficulty of the preparation method, reaching high animal numbers will be extremely challenging but perhaps possible in the future. Finally, together with the now available whole brain EM connectome, our data provide a timely resource for the community of fly neuroscientists interested in linking neuronal activity to behavior.

Using PCA/ICA as an unsupervised approach gave us insights into the organization of whole brain activity underlying brain states. This technique groups voxels that are correlated and thus misses more complex relationships. Furthermore, our method would not detect signals strongly corrupted by noise or different signals with the same spatial patterns. Nevertheless, it allowed us to extract components and compare their shape and localization to known structures. Based on these, we were able to propose a correspondence with specific candidate neurons (*Table 2*). Although some of these correspondences are very speculative, some components matched precisely the shape of known neurons (e.g., *Figure 5*), or reproduced the shape of the only neuron types present in the same region for specific Gal4 lines (e.g., MB compartments for dopamine *Aso et al., 2014*). Several of these neurons were previously implicated in walking or its modulation. For instance, EB–DA, found in data from dopaminergic neurons, shows among the strongest and most reliable activity during walk. These neurons were shown to be involved in ethanol-induced locomotion (*Kong et al., 2010*), are involved in sleep regulation (*Liang et al., 2016*), and were shown more recently to be involved in walk (*Fisher et al., 2022*), with a timing relative to walk onset consistent with *Figure 6*. Dopaminergic components in the MB also recapitulate previous studies (*Cohn et al., 2015*; *Siju et al., 2020*). Some correspondences between component and neurons are more speculative, such as IPS-Y and LAL-PS matching the shape of DNb02 and DNb01 neurons (*Table 2*). These neurons were found to project to the VNC (*Namiki et al., 2018*), and optogenetic activation of DNb01-induced twitching of the fly's front legs consistent with a role in turning (*Cande et al., 2018*). The component PB–EB is consistent with neurons in the previously described head direction cells circuit shown to receive movement-related information during navigation (*Lu et al., 2022*; *Lyu et al., 2022*; *Seelig and Jayaraman, 2015*). Among patterns not yet reported, the AVLPshell might be generated by a serotonergic neuron type of unknown function (e.g., Trh-F-100082; *Chiang et al., 2011*). Additional patterns such as the M-Omega pattern in datasets from dopaminergic cells might correspond to unknown neurons or a combination of tightly coupled neurons. Although most components were correlated with walk, others had an $R^2$ indistinguishable from zero thus likely representing ongoing activity unrelated to walk (*Schaffer et al., 2021*) or not captured by a linear model with a binary regressor for walk.

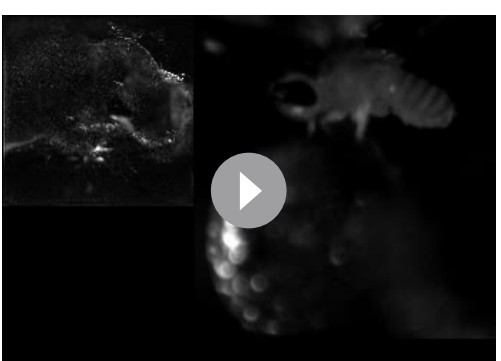

**Video 8.** Video showing whole brain activity of a fly with a severed connection between the brain and ventral nerve cord (VNC) being forced to walk on a treadmill. Note that the fly is still capable of walking on the rod when being forced. We did not, however, observe spontaneous walking activity.
https://elifesciences.org/articles/85202/figures#video8

Our data provide an entry point to relate anatomical connectivity to brain states and activity in brain-wide neural networks (*Aimon and Grunwald Kadow, 2019*). Whole brain connectomics in several small organisms including *Drosophila* has shown that neural networks extend across the entire brain with many pathways not predicted by single neuron or small motif imaging or manipulation of individual neurons (*Li et al., 2020*; *Scheffer et al., 2020*; *Zheng et al., 2018*). Whole brain imaging, as previously demonstrated in other model animals such as larval zebrafish, is a powerful method for observing these brain-wide networks to study their contribution to behavior and ultimately to local activities within specific neurons or brain regions. Whole brain data can be used to build functional connectomes allowing speculations about how information underpinning behavior travels throughout a whole brain. While such data can be generated for other animals including humans, the fly (along with *C. elegans* and in the near future zebrafish) currently provides the important advantage of being able to combine such activity maps with highly detailed anatomical maps from light microscopy with cellular resolution and whole brain EM connectomics with synaptic resolution. Ultimately, such data could be used to generate precise models of how recorded neural activity spreads through a brain. For instance, DNs receive input from regions of the brain that are innervated by outputs from higher brain regions such as the MB and CC (*Hsu and Bhandawat, 2016*). DN and AN innervation of the GNG is consistent with an important role of the GNG in motor control and motor feedback integration (*Chen et al., 2023*; *Tsubouchi et al., 2017*). How anatomical connectivity relates to activity in widespread neural networks is not clear. Our method and data complement some of the ongoing efforts to fill this significant gap in our knowledge.

## Walk elicits differential activities in neuromodulatory neurons

Perhaps not surprisingly, neuromodulatory systems participate and show signatures of ongoing behavior in the adult fly brain (see *Figure 3*). Dopaminergic and octopaminergic neurons are broadly activated when the fly walks compared to rest or groom (see *Figure 3*; *Aimon et al., 2019*; *Siju et al., 2020*), while serotonergic neurons show more complex activation patterns and timing with areas such as the AVLP being inhibited during walk (see *Figure 3* and *Figure 6—figure supplement 2*).

Our data are consistent with data from other species and suggest that relationships between the activity of neuromodulatory neurons and locomotion are broadly conserved across species. Dopaminergic neurons are activated during locomotion in mammals, with some neurons reporting ongoing behavior (*Howe and Dombeck, 2016*) while others show activity increase preceding behavior onset (*Coddington and Dudman, 2019*). We find that although the majority of fly dopamine neurons are activated during ongoing behavior, some dopaminergic neurons are activated hundreds of milliseconds before the onset of walk in some trials (e.g., WE-DA, M-Omega pattern, and γ1–3 compartments of the MB in some trials, *Figure 6*). Norepinephrine/octopamine is also a key neuromodulatory system involved in arousal in a variety of species (*Berridge, 2008*), with reports of an increase in activity of these neurons during locomotion in mammals (*Gray et al., 2021*; *Kaufman et al., 2020*; *Xiang et al., 2019*). In vertebrates, basal ganglia and brainstem aminergic neurons affect the cortico-basal ganglia–thalamic loops. A disruption of these loops can result in a loss of motor control (*Vicente et al., 2020*). Such loops likely exist in insects, too. For example, several octopaminergic neurons in the SEZ connect lower brain regions to the MB (*Busch et al., 2009*). Interestingly, serotonergic neurons show mixed roles in the control of motor behavior in different vertebrate species (*Dayan and Huys, 2015*; *Flaive et al., 2020*; *Vitrac and Benoit-Marand, 2017*), as well as in insect, where it regulates various types of motor behaviors including feeding, aggression and larval locomotion (e.g., *Aonuma, 2020*; *Helfrich-Förster, 2018*; *Ngai et al., 2019*; *Schoofs et al., 2018*; *Schoofs et al., 2014*). In particular, serotonin has been implicated in behavioral inhibition (e.g., patience; *Doya et al., 2021*) and as an opposing system to dopamine (*Dayan, 2012*). This is consistent with our observation that serotonergic neurons in some brain regions such as the AVLP are inhibited during walk (see also *Howard et al., 2019* for serotonin in the VNC). The AVLP receives input from ANs from the VNC conveying somatosensory information from the legs (*Tsubouchi et al., 2017*). Furthermore, calcium imaging revealed a spatial map for the AVLP and WED with neurons responding primarily to movement of fore-, mid-, or hind-legs (*Chen et al., 2023*; *Tsubouchi et al., 2017*), and the AVLP contains several descending neurons projecting toward the VNC (*Namiki et al., 2022*; *Namiki et al., 2018*). Serotonin inhibition in the AVLP could thus also be affecting a sensorimotor loop. However, we also found some serotonin-ergic neurons activated during walk, suggesting that, as for mammals, serotonin neurons are strongly

heterogeneous (*Dayan and Huys, 2015*). Our study thus suggests that key aspects of activity of neuromodulatory neurons during locomotion are conserved between mammals and insects despite their evolutionary distance.

## Origin of broad activation during ongoing behavior

Our results are most consistent with a model where proprioceptive, walking and leg sensory information are sent from the VNC into the GNG at the base of the fly's brain. Then, how is walk-related information relayed to the brain? Tuthill and colleagues recently identified some of the presumably many neural substrates in the VNC that receive, process and relay proprioceptive sensory information from the legs to the CNS (*Agrawal et al., 2020*; *Mamiya et al., 2018*). Their findings provide strong support for an important role of proprioception in movement and locomotion control in the adult fly. This information is transmitted by ANs from the VNC to the central brain. Less was known regarding the type, connectivity and function of the likely dozens or more AN in *Drosophila*, but screening approaches and advanced imaging techniques have shed light on some of them (*Allen et al., 2020*; *Sen et al., 2019*). In particular, a very recent study analyzed behavior-related activity of dozens of ANs labeled with specific Gal4 lines (*Chen et al., 2023*) and came to the complementary conclusion that ANs innervating regions such as the AVLP and GNG are poised to convey behavioral state and self-motion to several brain regions.

While grooming or forced walk on a treadmill does not require an active brain, lesions of the neck connectives as we have carried out dramatically decrease spontaneous walking in locusts (*Kien, 1990a*; *Kien, 1990b*), indicating that at least initiation of walk can be dependent on the brain. On the other hand, a cat with a severed spinal cord, like our fly with a severed VNC, maintains a highly coordinated walk pattern when forced to walk on a treadmill (*Afelt, 1974*). Thus, coordinated walk appears to mainly depend on central pattern generators in the spinal cord or VNC and is largely independent of brain input. Surprisingly, whole brain activity induced by spontaneous walking was similar to the activity we observed by forcing the animal to walk on a rotating rod (*Figure 7*). This result and our finding that the activity induced by walk in the GNG stems in a large part from axons are consistent with the interpretation that walk itself and not top-down motor control is responsible for the majority of activity observed in actively moving animals' brains. In the future, imaging whole brain activity while activating or silencing specific leg proprioceptive neurons or other ANs would confirm these findings and help dissect their contributions.

Once the walk signals reach the brain, neurons connecting lower to higher areas (e.g., neurons underlying the PENP-CL, GNG-SLP, and WPNb components, as well as octopaminergic neurons) could relay this information to higher areas. Interestingly, our data that inhibitory neurons are also broadly activated is inconsistent with the idea that broad brain activation arises due to a global disinhibition during periods of walking (*Benjamin et al., 2010*), although additional experiments will be necessary to ensure that activity in individual regions is not dominated by a small subset of activated neurons in a sea of inhibited neurons.

Importantly, however, we identified several brain components and small subregions, for instance in the posterior slope, that were activated hundreds of milliseconds before the fly started to walk (see *Figure 7*). This activity was delayed in forced walk and started only when the fly had started to move. This suggests a potential role of these areas in initiating walk. Activity before walk onset could also be due to preparatory movements that were not detected as walk (*Ache et al., 2019*), or represent the activity of neurons downstream of neurons responsible for triggering walk.

## Role of broad activation during ongoing behavior

One important concept to explain the role of behavioral state-dependent neural modulation is 'active sensation' (*Busse et al., 2017*). Essentially, ongoing movement can shape how neurons respond to visual, somatosensory, and possibly other sensory stimuli (*Chapman et al., 2018*; *Cruz et al., 2021*; *Fenk et al., 2021*; *Henschke et al., 2021*; *Wolpert et al., 2011*). For example, extracellular recordings from mouse V1 neurons walking on a ball showed that evoked visual responses differ in neurons of moving animals and those of still animals (*Dadarlat and Stryker, 2017*). Similarly, fly visual neurons respond more strongly to stimuli during walk or flight (*Chiappe et al., 2010*; *Maimon et al., 2010*; *Suver et al., 2016*). Our data indicate that the increase whole brain activity is elicited at walk onset and maintained afterwards (*Figure 6*). Since the overall increase in brain activity is not larger during

forced walk as compared to spontaneous walk, our data also suggest that global brain activity does not generally represent a mismatch, or error signal, between actual and predicted proprioceptive feedback during walk. These observations support the conclusion that movement specific information reach the brain and modulate brain activity widely. Such information could serve multiple purposes from uncoupling of sensory-to-motor information, that is own movement vs. movement of environment, to learning of complex movements (*Lu et al., 2022*).

## Conclusions

We provide an overview of global brain activity during simple behaviors in *Drosophila*. As for other animals, *Drosophila* brain activity is globally correlated with locomotion representing a global change in brain state. However, our results challenge the assumption that most of the activity is related to decision-making, top-down motor control, or prediction error detection from sensory feedback and instead suggest that walk itself and somatosensory bottom-up stimuli are largely responsible. By using a combination of pan-neuronal and specific neuron subtype imaging, we shed light on the brain location and nature of neurons that respond so strongly to behavior. Altogether, our data provide a novel resource for generating new hypotheses regarding the brain-behavior loop and for dissecting the neural circuits and computations underpinning it.

# Materials and methods

**Key resources table**

| Reagent type (species) or resource | Designation | Source or reference | Identifiers | Additional information |
|---|---|---|---|---|
| Genetic reagent (*D. melanogaster*) | UAS-GCaMP6s | Bloomington *Drosophila* Stock Center | BDSC_42749 | |
| Genetic reagent (*D. melanogaster*) | UAS-GCaMP6m | Bloomington *Drosophila* Stock Center | BDSC_42750 | |
| Genetic reagent (*D. melanogaster*) | UAS-GCaMP7s | Bloomington *Drosophila* Stock Center | BDSC_79032 | |
| Genetic reagent (*D. melanogaster*) | UAS-GCaMP6f | Bloomington *Drosophila* Stock Center | BDSC_42747 | |
| Genetic reagent (*D. melanogaster*) | UAS-GCaMP7f | Bloomington *Drosophila* Stock Center | BDSC_79031 | |
| Genetic reagent (*D. melanogaster*) | UAS-syt-GCaMP6s | Vanessa Ruta | N/A | |
| Genetic reagent (*D. melanogaster*) | GMR58E04-Gal4 | Bloomington *Drosophila* Stock Center | BDSC_41347 | |
| Genetic reagent (*D. melanogaster*) | TH-Gal4 | Bloomington *Drosophila* Stock Center | BDSC_8848 | |
| Genetic reagent (*D. melanogaster*) | Tdc2-Gal4 | Bloomington *Drosophila* Stock Center | BDSC_9313 | |
| Genetic reagent (*D. melanogaster*) | Trh-Gal4 | Bloomington *Drosophila* Stock Center | BDSC_38388 | |
| Genetic reagent (*D. melanogaster*) | Ddc-Gal4 | Bloomington *Drosophila* Stock Center | BDSC_7009 | |
| Genetic reagent (*D. melanogaster*) | TH, 58E02-Gal4 | Siju et al. | N/A | |
| Genetic reagent (*D. melanogaster*) | TH, DDC-Gal4 | This paper | N/A | |
| Genetic reagent (*D. melanogaster*) | Vglut-Gal4 | Bloomington *Drosophila* Stock Center | BDSC_24635 | |
| Genetic reagent (*D. melanogaster*) | Cha-Gal4 | Bloomington *Drosophila* Stock Center | BDSC_6798 | |
| Genetic reagent (*D. melanogaster*) | Nsyb-Gal4 | Bloomington *Drosophila* Stock Center | BDSC_51635 | |
| Genetic reagent (*D. melanogaster*) | GMR57C10-Gal4 | Bloomington *Drosophila* Stock Center | BDSC_39171 | |
| Genetic reagent (*D. melanogaster*) | Gad1-Gal4 | Bloomington *Drosophila* Stock Center | BDSC_51630 | |
| Software, algorithm | Python 3 | Python Software Foundation | https://www.python.org | |
| Software, algorithm | ImageJ/FIJI | *Schindelin et al., 2012* | https://fiji.sc/ | |
| Software, algorithm | MATLAB | MATLAB | mathworks.com | |
| Software, algorithm | Analysis code | GitHub | https://github.com/sophie63/Aimon2022 | |

### Fly preparation for imaging

We used 1- to 4-day-old female flies raised at 25°C. Most flies were starved 24 or 48 hr with a water only environment, and we clipped their wings at least 1 day in advance. Experiments were performed in the evening peak of circadian activity (ZT0 or ZT11) and we heated the room to ~28°C during the experiment. In total, we recorded brain activity and behavior from 84 adult female flies.

We prepared the flies as described in detail in *Woller et al., 2021*. Briefly, we fixed a fly to a custom-designed 3D printed holder, so as to allow access to the whole posterior side of the head while the legs were free to move. We added saline (103 mM NaCl, 3 mM KCl, 5 mM TES (N-Tris(hydroxymethyl)methyl-2-aminoethanesulfonic acid)).

### Walk substrates

For studying spontaneous walk, we used two types of small balls. One was an air-supported ball as previously described (*Sayin et al., 2019*). As we wanted to make sure the walk was initiated by the fly rather than erratic movement of the ball, we also used small Styrofoam balls that were held by the fly. The speed of the rotational flow, which is proportional to the degree of turning, was 0.4–2 rad/s.

For the treadmill to study forced walk and turn, we used small motors (DC 6V gear motor with long M3 × 55 mm lead screw thread output shaft speed reducer Walfront Store, https://www.amazon.de/), covered with self-curing rubber (Sugru by tesa) to provide a smoother surface. The speed for forced walk was between 1.5 and 6 mm/s. The rotational speed for forced turn was 0.3–2 rad/s.

### In vivo light field imaging

Fast volumetric imaging was performed using light field imaging – in which a microlens array separates rays from different angles to give information on depth – and was carried out as previously described in detail by *Aimon et al., 2019*. A few datasets were previously published in *Aimon et al., 2019* and source data (http://dx.doi.org/10.6080/K01J97ZN), with a microscope equipped with a ×20 NA1.0 objective. Most data were obtained with a light field microscope constituted of a Thorlabs Cerna system with a Leica HC FLUOTAR L ×25/0.95 objective and an MLA-S125-f12 microlens array (Viavi). The microlens array was placed on the image plane, while the camera imaged the microlens array through 50 mm f/1.4 NIKKOR-S Nikon relay lenses. The light field images were recorded with a scientific CMOS camera (Hamamatsu ORCA-Flash 4.0). The volumes were reconstructed offline, using a python program developed by *Broxton et al., 2013* and available on github: https://github.com/sophie63/FlyLFM (copy archived at *Aimon, 2023b*).

Given that the maximum recording speed depended on the expression of the Gal4-line, the UAS-reporter, and each individual fly preparation, that is the limiting factor was the signal-to-noise ratio for lines with low expression, we started recording at 2 or 5 Hz for the first experiment. If the quality of recording suggested that higher speeds were possible, we increased recording speed to a maximum of 98 Hz. If the viability of the fly allowed for it, we recorded experiments on air-supported and Styrofoam balls in the same fly.

### Behavior recording and scoring

We imaged the fly and substrate movements using infrared illumination and two small cameras (FFMV-03M2M from Point Grey) triggered by the fluorescence recording camera to ensure temporal alignment between fluorescence and behavior.

Walking, flailing, and grooming were obtained by measuring the optic flow from the Movies of the ball or by analyzing the movement of the fly's legs using the 'optic flow' plugin in FIJI. For turning or rotational speed (rad/s), the sum of left or right optic flows was not binarized. For regression of neuronal time series with behavior, all behavioral time series were then convolved with the single spike response of the GCaMP version used for the experiment and subjected to the same $\Delta F/F$ procedure as the fluorescence time series (see below).

### Pre-processing

Reconstructed volumetric fluorescence data were pre-processed by first correcting for movement using 3Dvolreg from AFNI (https://github.com/afni/afni) (*Cox, 1996*; *Cox and Hyde, 1997*). In Matlab, we then calculated the $\Delta F/F$ for each voxel by subtracting and dividing by the signal averaged for

4000 time points. We finally decreased noise with a Kalman filter (from https://www.mathworks.com/matlabcentral/fileexchange/26334-kalman-filter-for-noisy-movies) with a gain of 0.5.

| Gal4 | UAS-GCaMP | # of flies, spontaneous Walk (flail, groom) | Flailing | Grooming | Frame rate in Hz (# of experiments) | Walking substrate (# of experiments) | # flies of flies forced walk |
|---|---|---|---|---|---|---|---|
| Total | Total | 58 | 9 | 6 | 98 (8), 50 (21), 20 (11), 10 (9), 5 (30), 2 (2) | Air-supported ball (33), styrofoam ball (54) | 26 |
| Nsyb | 7f | 1 | 1 | 1 | 50 (1) | Air-supported ball | |
| | 7s | 1 | 1 | | 5 (1) | Styrofoam ball | 1 |
| | 6m | 6 | 5 | 2 | 50 (8), 20 (5), 10 (3), 5 (4) | Air-supported ball (16), styrofoam ball (5) | 3 |
| | 6f | 3 | | 1 | 98 (2), 50 (1) | Air-supported ball (2), styrofoam ball (1) | |
| | 6s | 1 | 1 | | 50 (1) | Styrofoam ball (1) | 2 |
| | Total | 12 | 8 | 4 | 98 (2), 50 (11), 20 (5), 10 (3), 5 (5) | Air-supported ball (19), styrofoam ball (8) | 6 |
| R57C10 | 7s | 1 | | 1 | 98 (2) | Styrofoam ball (2) | |
| | 6s | 3 | 1 | 1 | 50 (1), 5 (1), 20 (1) | Styrofoam ball (3) | 3 |
| | Total | 4 | 1 | 2 | 98 (2), 50 (1), 20 (1), 5 (1) | Air-supported ball (0), styrofoam ball (5) | 3 |
| Cha | 6m | 4 | | | 5 (4) | Styrofoam ball (4) | |
| | 6f | 1 | | | 98 (1) | Air-supported ball (1) | |
| | Total | 5 | | | 98 (1), 5 (4) | Air-supported ball (1), styrofoam ball (4) | 0 |
| Vglut | 6m | 5 | | | 20 (2), 10 (3), 5 (4) | Air-supported ball (3), styrofoam ball (6) | |
| | Total | 5 | | | 20 (2), 10 (3), 5 (4) | Air-supported ball (3), styrofoam ball (6) | 0 |
| Gad | 6m | 5 | | | 98 (2), 20 (1), 5 (5), 2 (1) | Styrofoam ball (9) | |
| | Total | 5 | | | 98 (2), 20 (1), 5 (5), 2 (1) | Air-supported ball (0), styrofoam ball (9) | 0 |
| TH/DDC | 6m | 5 | | | 50 (2), 5 (1), 10 (1) | Air supported ball (3), styrofoam ball (2) | 4 |
| | 6f | 6 | | | 50 (5), 98 (1) | Air supported ball (2), styrofoam ball (3) | 1 |
| | Total | 11 | | | 98 (1), 50 (7), 10 (1), 5 (1) | Air-supported ball (5), styrofoam ball (5) | 5 |
| Trh | 6m | 3 | | | 50 (3), 5 (3) | Air supported ball (1), styrofoam ball (6) | |
| | 6f | 3 | | | 10 (2) | Air supported ball (2), styrofoam ball (1) | |
| | 6s | 3 | | | 5 (3) | Styrofoam ball (4) | 6 |
| | Total | 9 | | | 50 (3), 10 (2), 5 (6) | Air-supported ball (3), styrofoam ball (11) | 6 |
| Tdc2 | 6s | 7 | | | 20 (2), 5 (4), 2 (1) | Air supported ball (2), styrofoam ball (6) | 6 |
| | Total | 7 | | | 20 (2), 5 (4), 2 (1) | Air-supported ball (2), styrofoam ball (6) | 6 |

We generated summary movies by maximum projecting the $\Delta F/F$ volumes and combining these to the behavior.

## Alignment to template

We aligned the functional data to the anatomical template JRC2018 (https://www.janelia.org/open-science/jrc-2018-brain-templates) using the landmarks registration plugin with ImageJ (as described in http://imagej.net/Name_Landmarks_and_Register), with the landmarks found in https://github.com/sophie63/Aimon2022/blob/main/Registration/SmallJRC2018Template.points.

This allowed for extracting the anatomically defined regions covered by the functional regions (see *Table 2*) and finding candidate neurons using Flycircuit (http://www.flycircuit.tw/) or Virtual Fly Brain https://v2.virtualflybrain.org databases.

## Statistical analysis

Sample size determination: As we expected large effects (*Aimon et al., 2019*), we chose to focus on those and planned for a minimum of 5 flies per condition satisfying quality criteria. Low-quality flies were flies in which no spontaneous activity was detected, or in which front legs were not touching the substrate for walking experiments.

Statistics were performed in python with code freely available on https://github.com/sophie63/Aimon2022. (*Aimon, 2023a* copy archived at swh:1:rev:a6499a918cc1373db3933a061c6ad-

d7c57b79cf2). To compare fluorescence time series (normalized by the absolute maximum value per fly) and behavioral time series, we used a simple regression model: $\frac{\Delta F}{F} = f\left(behavior\ regressor\right)$, solved with the ordinary least square fit function of the python statsmodels package. For each time series (either regional averaged intensity or the PCA/ICA component), this provided a fraction of variance explained by the behavior ($R^2$), and the sign and strength of the correlation (coefficient). We compared these values with pairwise tests using two-sided Mann–Whitney non-parametric tests with a Bonferroni multiple comparison correction. We used a more complex linear model to evaluate the effect of variables of interest (behavior, brain region, neural type) while explaining confounds (GCaMP version, exact pan-neuronal Gal4): $R^2$ ~ Behavior + RegionNames + Gal4 + UAS and Coef ~ Behavior + RegionNames + Gal4 + UAS. We then plotted the coefficients + intercept, and 95% interval of coefficient + 95% interval of the intercept to compare the effect of the variables to zero. We also compared these values with zero with a $t$-test and corrected for multiple comparison using the Benjamini–Hochberg method. For detection of activation before walk, we integrated activity from −0.5 s before to walk onset minus level at −0.5 s before onset. We used a Wilcoxon test to compare these values to zero and corrected for multiple comparison using the Benjamini–Hochberg method. Flies were recorded over prolonged periods of times at different imaging speeds and during different behaviors. See table above for details on number of individual flies and types of experiments per fly.

## PCA/ICA

To obtain functional maps of the fly brain, we performed PCA and ICA as described previously (*Aimon et al., 2019*; *Beckmann and Smith, 2004*). Briefly, SVD (singular value decomposition) was used a first time to find the level of noise and normalize voxels by their noise variance. SVD was performed a second time on this normalized data resulting in maps and time series for principal components. The principal component maps were unmixed using ICA to obtain localized regions. The same matrix was used to unmix the time series.

## Acknowledgements

We are very grateful to Marta Costa and Kei Ito for sharing data, images, and knowledge during the course of this study. We also thank Francisco Rodriguez-Jimenez, Paul Bandow, Subhadarshini Parhi, and Kunhi Purayil Siju for help with data analysis. We would like to acknowledge funding from the European research commission (ERCStG FlyContext to IGK, ERCStG NeuroDevo to JG), the Simons Foundation (Aimon - 414701 to SA), and the Ministry of Culture and Science of the State of North Rhine-Westphalia (iBehave network to IGK).

## Additional information

### Competing interests

Ilona C Grunwald Kadow: Reviewing editor, eLife. The other authors declare that no competing interests exist.

### Funding

| Funder | Grant reference number | Author |
|---|---|---|
| European Research Council | ERCStG FlyContext | Ilona C Grunwald Kadow |
| European Research Council | ERCStG NeuroDevo | Julijana Gjorgjieva |
| Simons Foundation | Aimon - 414701 | Sophie Aimon |
| iBehave network funded by the Ministry of Culture and Science of the State of North Rhine-Westphalia | | Ilona C Grunwald Kadow |

| Funder | Grant reference number | Author |
|---|---|---|

The funders had no role in study design, data collection, and interpretation, or the decision to submit the work for publication.

## Author contributions

Sophie Aimon, Conceptualization, Data curation, Software, Formal analysis, Validation, Investigation, Visualization, Methodology, Writing – original draft, Writing – review and editing; Karen Y Cheng, Data curation, Validation, Investigation, Writing – review and editing; Julijana Gjorgjieva, Supervision, Project administration, Writing – review and editing; Ilona C Grunwald Kadow, Conceptualization, Resources, Supervision, Funding acquisition, Investigation, Visualization, Methodology, Writing – original draft, Project administration, Writing – review and editing

## Author ORCIDs

Sophie Aimon http://orcid.org/0000-0002-0990-0342
Karen Y Cheng http://orcid.org/0000-0002-4256-1688
Julijana Gjorgjieva http://orcid.org/0000-0001-7118-4079
Ilona C Grunwald Kadow http://orcid.org/0000-0002-9085-4274

## Decision letter and Author response

Decision letter https://doi.org/10.7554/eLife.85202.sa1
Author response https://doi.org/10.7554/eLife.85202.sa2

# Additional files

## Supplementary files

- MDAR checklist
- Supplementary file 1. Description of components and putative underlying neurons.

## Data availability

Time series of regional data are available on Dryad https://doi.org/10.5061/dryad.3bk3j9kpb, and small datasets of processed data used for generating figures are on github: https://github.com/sophie63/Aimon2022. Code to analyze the data is available on https://github.com/sophie63/Aimon2022 (copy archived at *Aimon, 2023a*) and https://github.com/sophie63/FlyLFM (copy archived at *Aimon, 2023b*). Original data are very large (several tens of TB) and is available upon request to Ilona.grunwald@uni-bonn.de.

The following dataset was generated:

| Author(s) | Year | Dataset title | Dataset URL | Database and Identifier |
|---|---|---|---|---|
| Aimon S, Cheng KY, Gjorgjieva J, Ilona C GK | 2022 | Time series of regional activity in *Drosophila* | https://doi.org/10.5061/dryad.3bk3j9kpb | Dryad Digital Repository, 10.5061/dryad.3bk3j9kpb |

The following previously published dataset was used:

| Author(s) | Year | Dataset title | Dataset URL | Database and Identifier |
|---|---|---|---|---|
| Aimon S, Katsuki T, Jia T, Grosenick L, Broxton M, Deisseroth K, Sejnowski TJ, Greenspan RJ | 2019 | Whole-brain recordings of adult *Drosophila* using light field microscopy along with corresponding behavior or stimuli data | https://doi.org/10.6080/K01J97ZN | Collaborative Research in Computational Neuroscience, 10.6080/K01J97ZN |

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
