## [Editor Report]

This paper expands on prior work by using whole-brain calcium imaging in *Drosophila* to examine how spontaneous and forced walking and turning affect neural activity in the brain. The measurements presented will serve as a valuable resource for the fly systems neuroscience community and suggest many testable hypotheses that may serve as the basis for future studies. Analyses of the data are solid, and presented with appropriate caveats. This article will be of interest to neuroscientists engaged with the central problem of how behavior modulates neural activity.

---

## [Decision Letter]

**Decision letter after peer review:**

Thank you for submitting your article "Combined patterns of activity of major neuronal classes underpin a global change in brain state during spontaneous and forced walk in *Drosophila*" for consideration by *eLife*. Your article has been reviewed by 3 peer reviewers, one of whom is a member of our Board of Reviewing Editors, and the evaluation has been overseen by Claude Desplan as the Senior Editor. The reviewers have opted to remain anonymous.

Essential revisions:

The reviewers have agreed that the following points should be addressed in a revised manuscript. The list here is short but applies widely to many parts of the manuscript, and many (but not all) instances of applications may be found in the reviewer's individual comments. Especially item (1) below – tempering claims – seems to require a substantial revision of the claims and the prose, i.e., not just adding in a few hedging words.

1) The authors should adjust and temper their claims to match the evidence presented. Reviewers were particularly worried about claims of causality, claims of correspondence with individual neurons, and lack of nuance in interpreting neuropil averages over many neurons. Comments in each review point out specific cases.

2) The statistical rigor could be improved and should account for multiple comparisons. In data analysis, caveats about patterns of responses and metrics should be made clear. Several specific issues are pointed out in individual reviews.

3) The reviewers agreed that the paper could be profitably shortened by focusing the text more tightly on conclusions and avoiding speculation, especially in the results.

*Reviewer #1 (Recommendations for the authors):*

1. A major metric in this paper is the correlation coefficient squared for dF/F and walking in each brain region. This has some important potential confounds that don't seem to get enough airtime in the analysis or interpretation of results. In particular, this metric is particularly susceptible to signal-to-noise of the data. This means that differences in this metric between brain regions could be due to simple differences in relative expression (and thus brightness and SNR) between regions. Similarly, and perhaps more importantly, when using this metric to make statements about neuron types using different neurotransmitters or amines, the different strengths and patterns of the drivers could be creating the differences in patterns. Or the differences could all be real, but it is difficult to evaluate with the data provided and analyzed as shown.

2. The regression model for predicting dF/F for each region from a walking binary variable was interesting. It would be important to account for multiple comparisons here to identify those regions one is most confident are non-zero. And I think this analysis could also be used productively to compare patterns between conditions – the cosine angle between the vector of regressors tells you succinctly about how different patterns are. This could be used between conditions and between experiments within conditions.

3. The averaging of neuropils in the analyses in this paper has given the authors some valuable insights into patterns of activation across the brain. However, it also hides a lot of information. For instance, identical average activity in a region could correspond to very different activations of individual neurons within the region. Similarly, differences in regional activation between different Gal4s could be due simply to non-uniform expression among different subtypes that results in different dF/F signals. One example is the interpretation of the GAD and vglut drivers, where the activity tied to walking is used to conclude that the activity in the brain is not due to generalized disinhibition. But we have no idea whether the activity measured in these experiments represents 1% of neurons becoming more active and 99% being hyperpolarized (potentially invisible with Ca indicators), or whether it represents 99% of neurons becoming activated. This is an example of where I expect the authors' interpretation is likely to be correct, but their conclusions are stated more strongly than the data permits. This is just one example, but I think it applies to most analyses in the paper. Throughout, this averaging over neuropil can be masking a lot, so it's rather hard to interpret in some of the simple ways that the authors do. I think the caveats with tying these measurements to individual neurons or patterns of activity should be given more discussion and featured heavily in the results and potential interpretations.

4. I think that tabulating potential neurons of interest in each neuropil are a valuable resource, but it seems to be presented as far too strong a conclusion, when in fact it is quite speculative. To show this with the certainty suggested by the current prose, I think one would need to show that expression in just those neurons reproduces the response pattern, or that the pattern is changed without those neurons. As it stands, I think substantially more restraint is required in the discussion of these speculations.

5. The comparisons between spontaneous and forced walking are the only manipulations in the paper and provide interesting results about neuropils active before spontaneous walking. However, the authors repeatedly refer to this as causal – that those regions are triggering the walking (in the discussion). (They say 'likely' once, but thereafter seem to treat this as factual.) This seems too strong – that causality is speculation for the discussion, not a concluded fact from these experiments. The region activities are correlated with and precede walking (and could even be predictive in impending walking – that could be an interesting analysis!), but even then, without some manipulation of those neurons or regions, it doesn't seem possible to conclude that they trigger walking, rather than just precede it. (Alternative hypothesis, not ruled out by this observational data: single neuron pair somewhere, roughly invisible in the averaging done here, elicits spontaneous walking and also activates the neuropils the authors observe.)

6. The asymmetric activity patterns under turning are a cool result. Here also, I think it would be appropriate to exercise more caution in assigning this to specific neurons.

7. Overall, I think the length of some sections in the results diffuses the points the authors want to make. Editing to make it more concise in the results, especially, removing speculation and labeling it more explicitly as speculation in the discussion, would result in a clearer paper.

*Reviewer #2 (Recommendations for the authors):*

Results:

First paragraph: "We expressed GCaMP pan-neuronally (i.e., *nsyb*-gal4;UAS-GCaMP6s/6m/6f/7s/7f)". The text in the parens is confusing because it does not help the reader understand when different versions of GCaMP were used and why.

Second paragraph: Movie 1-can you comment on the extent to which the presence of the esophagus obscures activity in the subesophageal zone?

Third paragraph: The idea that frame rates between 5-98 Hz are not different for this analysis is surprising. Can you comment on why the frame rate does not seem especially relevant for these experiments? Is this due to the speed of GCaMP6f?

Fifth paragraph: Can you expand on the effect of the version of GCaMP?

Aimon et al. 2019 used norpA mutants to show that global increases in brain activity during walk are independent of visual sensory input. Can you comment on why you repeated this experiment with black nail polish? How do these results enrich our understanding beyond what you have previously published?

Figure S1: Please provide appropriate citations for Figure S1, panel K, and indicate that this image has been reproduced from another publication.

Sixth paragraph: The difference in R2 for global brain activation for different walking substrates seems important but is somewhat glossed over. Can you expand on why you think different walking substrates impact R2 for global brain activation?

Both inhibitory and excitatory neurons are recruited during walk.

First paragraph: The way this paragraph is written, the reader is led to believe that you are looking for specific descending neuron types and investigating their neurontransmitter profiles. Consider rephrasing to convey that you are looking at all neuronal classes (interneurons, ascending neurons, descending neurons) and investigating how neurons that release different types of neurotransmitters contribute to global brain activity during walking.

Second paragraph: Why change to GCaMP6m here after using GCaMP6f for the global experiments?

Figure 2A. Why do you think the R2 during walk for Cha, VGlut, and GAD is higher than it is for pan-neuronal drivers? Is the relatively high R2 for GAD sufficient to conclude that GABAergic neurons explain relatively more of the variance during walk?

Aminergic neuron activity is strongly correlated with behavior

Figure 3. To what extent are activity maps for more sparse neuromodulatory neuron types related to the expression patterns of these lines? If you were to activate all neurons in these lines with something like Chrimson (rather than with walking), would you see similar or different subsets of regions activated?

Whole brain activity data identifies specific brain regions or even neurons.

Fifth paragraph:

"Although most component were correlated with walk, other had an R2 indistinguishable from zero thus likely representing ongoing activity unrelated to walk (Schaffer et al., 2021)." Revise to "Although most components were correlated with walk, others had an R2 that was indistinguishable from zero thus likely representing ongoing activity unrelated to walk (Schaffer et al., 2021)."

Turning activates specific brain regions and neurons.

First paragraph:

"These DNs were previously described by Namiki et al., who generated specific Gal4 lines for neuronal manipulation (Namiki et al., 2018; Robie et al., 2017)." Aren't these split-GAL4 lines?

Many neurons have similar morphology, so concluding that activity in IPS-Y and LAL-PS is due to specific DNs should await further experiments. For example, subtracting DNb02 and DNb01 from the GAL4 pattern with LexA expressing GAL80 to show this IPS-Y and LAL-PS activity disappears would be helpful. Alternatively, the authors could image DNb02 and DNb01 split-GAL4 lines to show comparable dynamics of activation during turning.

"As for forward walk, most components could be matched to candidate neurons (Table 2) for future functional analysis." Generally speaking, it would be helpful to communicate more strongly that any component to candidate neuron matching is tentative. The data presented here is insufficient to conclude that the components can be matched with specific neuronal cell types.

Brain dynamics at transitions between rest and walk.

Figure 6. Adding a y-axis scale (at least at the beginning of each row of graphs) would help to orient the reader. Additionally, more description of this figure in the figure legend is needed, especially to add that the red bar is presumably indicating walk onset?

Increases in activity before walk onset in PRW, posterior slope (IPS-Y, SPS), and GNGm are small and difficult to appreciate. Is there a more rigorous way to quantify the onset of activation, perhaps by looking at the derivative of normalized dF/F?

"Of note, the component PB-EB corresponds to the previously described head direction cells shown to receive movement-related information during navigation (Lu et al., 2022; Lyu et al., 2022; Seelig and Jayaraman, 2015). EB-DA, found in data from dopaminergic neurons, shows among the strongest and most reliable activity during walk. These neurons were shown to be involved in ethanol-induced locomotion (Kong et al., 2010) and are also involved in sleep regulation (Liang et al., 2016)."

How sure are the authors that the PB-EB component corresponds to head direction cells? I would suggest softening this assertion to express some uncertainty unless further confirmation experiments are added.

"Several other components displayed more variable, and at times too variable dynamics between flies to detect a clear direction (Figure S6)." Which components are the authors referring to? It would be helpful to elaborate on this either in the text or in the Figure S6 legend. For Figure S6, what does the grey vertical bar indicate in each plot?

"The observed activity patterns confirm and extend published neuronal manipulation data, where available, on sleep and locomotion (see above and discussion)" I would soften this claim to say that these data support published neuronal manipulation data. I do not think these data extend published neuronal manipulation studies as no such manipulations were performed. The data presented here could be used to generate hypotheses for future studies.

Forced walk and forced turning recapitulates most activity

Second paragraph: Is the expression of syt-GCAMP stronger in GNG, AMMC, and AVLP (this could be possible if there are more synapses in these regions)? If you activated the brain pan-neuronally with Chrimson and imaged using syt-GCaMP, would you see a different pattern of activity than what is seen suiting walk?

For Figure 7G/ the difference between spontaneous and forced walk, it would be helpful to add statistical comparisons of the normalized activity traces when claiming that they are different versus the same.

Discussion:

Role and origin of broad activation during ongoing behavior

First paragraph: Delete the underscore between "coupling_sensory"

Role and origin of broad activation during ongoing behavior

Fifth paragraph: "More generally, our data indicate that regions triggering walk are also highly sensitive to walk itself suggesting a closed-loop between walk initiation and walk maintenance, possibly including factors such as speed, detailed movement etc." The data presented here is insufficient to conclude that these regions are responsible for triggering walk without further neuronal manipulation studies. The authors should support this claim by showing that MB components correspond to specific neuronal cell types (by removing specific neurons from the global pattern or imaging specific neurons and showing that they show identical dynamics) and then activate and silence these specifical neuronal cell types while analyzing walking behavior.

*Reviewer #3 (Recommendations for the authors):*

1) The word "underpin" in the title may be too strong because it can be construed to imply a degree of causality, which is not shown. I suggest more precise phrasing like "correlates with" or something similar that clearly does not claim causal relations.

2) Throughout the text activity correlations of many dozens of subregions are compared. E.g. Figure 3F tests 40 hypotheses. Figure 5b tests closer to 90. With so many brain regions it will be important to leverage a statistical framework that explicitly considers multiple hypothesis testing in order to better convey to a reader the relative confidence in each finding. Some avenues to consider: a q-value analysis could be used that articulates a false-discovery rate. Shuffled datasets can also be used to try to reject null hypotheses. And GFP-only animals that lack GCaMP could also be used to generate null distributions to compare against and to reject null hypotheses.

3) Figure S7: the gray shading and blue traces are not defined. At first, I had assumed gray is an error bar and blue is average, but if that's the case, there appears to be a major problem in Figure S7C where these diverge dramatically, e.g. in SMB and SPS and NO.

---

## [Author Response]

Essential revisions:The reviewers have agreed that the following points should be addressed in a revised manuscript. The list here is short but applies widely to many parts of the manuscript, and many (but not all) instances of applications may be found in the reviewer's individual comments. Especially item (1) below – tempering claims – seems to require a substantial revision of the claims and the prose, i.e., not just adding in a few hedging words.1) The authors should adjust and temper their claims to match the evidence presented. Reviewers were particularly worried about claims of causality, claims of correspondence with individual neurons, and lack of nuance in interpreting neuropil averages over many neurons. Comments in each review point out specific cases.

We agree with your assessment and have softened claims of correspondence with specific neurons, removed claims of causality from the result section, and added nuance in interpreting the neuropile data all through the manuscript.

2) The statistical rigor could be improved and should account for multiple comparisons.

We agree with your assessment and have softened claims of correspondence with specific neurons, removed claims of causality from the result section, and added nuance in interpreting the neuropile data all through the manuscript.

In data analysis, caveats about patterns of responses and metrics should be made clear. Several specific issues are pointed out in individual reviews.

We have made these caveats clearer throughout the manuscript.

3) The reviewers agreed that the paper could be profitably shortened by focusing the text more tightly on conclusions and avoiding speculation, especially in the results.

We have substantially shortened the result section (from 9 to 6 pages).

Reviewer #1 (Recommendations for the authors):1. A major metric in this paper is the correlation coefficient squared for dF/F and walking in each brain region. This has some important potential confounds that don't seem to get enough airtime in the analysis or interpretation of results. In particular, this metric is particularly susceptible to signal-to-noise of the data. This means that differences in this metric between brain regions could be due to simple differences in relative expression (and thus brightness and SNR) between regions. Similarly, and perhaps more importantly, when using this metric to make statements about neuron types using different neurotransmitters or amines, the different strengths and patterns of the drivers could be creating the differences in patterns. Or the differences could all be real, but it is difficult to evaluate with the data provided and analyzed as shown.

The reviewer raises the important point that differences in signal-to-noise ratio could explain or influence the differences in R^2^ values. The coefficient of the regression between δ F/F fluorescence and a regressor as in Figure 2C and 3B shouldn’t be affected by such a problem though as the normalization factor for the time series of regional activity was calculated globally for the whole brain. Indeed, a lower signal to noise ratio would affect the accuracy but not the value of the coefficient. We have rewritten the parts on R^2^ and explained the relationship to signal-to-noise ratios better.

2. The regression model for predicting dF/F for each region from a walking binary variable was interesting. It would be important to account for multiple comparisons here to identify those regions one is most confident are non-zero. And I think this analysis could also be used productively to compare patterns between conditions – the cosine angle between the vector of regressors tells you succinctly about how different patterns are. This could be used between conditions and between experiments within conditions.

We have added tests with multiple comparison correction for all multiple regions analysis and indicated the significant difference from zero as full markers.

We thank the reviewer for suggesting the use of cosine similarity to quantify the similarity of patterns across regions. We have added this analysis.

3. The averaging of neuropils in the analyses in this paper has given the authors some valuable insights into patterns of activation across the brain. However, it also hides a lot of information. For instance, identical average activity in a region could correspond to very different activations of individual neurons within the region. Similarly, differences in regional activation between different Gal4s could be due simply to non-uniform expression among different subtypes that results in different dF/F signals. One example is the interpretation of the GAD and vglut drivers, where the activity tied to walking is used to conclude that the activity in the brain is not due to generalized disinhibition. But we have no idea whether the activity measured in these experiments represents 1% of neurons becoming more active and 99% being hyperpolarized (potentially invisible with Ca indicators), or whether it represents 99% of neurons becoming activated. This is an example of where I expect the authors' interpretation is likely to be correct, but their conclusions are stated more strongly than the data permits. This is just one example, but I think it applies to most analyses in the paper. Throughout, this averaging over neuropil can be masking a lot, so it's rather hard to interpret in some of the simple ways that the authors do. I think the caveats with tying these measurements to individual neurons or patterns of activity should be given more discussion and featured heavily in the results and potential interpretations.

We have removed the conclusion that activation is not simply due to global disinhibition from the result section and have instead moved this idea to the discussion, with additional caution:

“Interestingly, our data that inhibitory neurons are also broadly activated is inconsistent with the idea that broad brain activation arises due to a global disinhibition during periods of walking (Benjamin, 2010), although additional experiments will be necessary to ensure that activity in the different regions is not dominated by a small subset of activated neurons in a sea of inhibited neurons. “

In addition, we added more nuance in interpretation of regional activity throughout the paper.

4. I think that tabulating potential neurons of interest in each neuropil are a valuable resource, but it seems to be presented as far too strong a conclusion, when in fact it is quite speculative. To show this with the certainty suggested by the current prose, I think one would need to show that expression in just those neurons reproduces the response pattern, or that the pattern is changed without those neurons. As it stands, I think substantially more restraint is required in the discussion of these speculations.

We agree with the reviewer that the correspondence to specific neurons was overstated. Except for a few examples where the neurons we propose are the only known neurons to be GAL4-expressing in one specific region, the correspondence for most components is indeed speculative. We have made this point clearer throughout the paper, but we still believe that naming candidate neurons is helpful to scientists working on motor behavior control and feedback.

5. The comparisons between spontaneous and forced walking are the only manipulations in the paper and provide interesting results about neuropils active before spontaneous walking. However, the authors repeatedly refer to this as causal – that those regions are triggering the walking (in the discussion). (They say 'likely' once, but thereafter seem to treat this as factual.) This seems too strong – that causality is speculation for the discussion, not a concluded fact from these experiments. The region activities are correlated with and precede walking (and could even be predictive in impending walking – that could be an interesting analysis!), but even then, without some manipulation of those neurons or regions, it doesn't seem possible to conclude that they trigger walking, rather than just precede it. (Alternative hypothesis, not ruled out by this observational data: single neuron pair somewhere, roughly invisible in the averaging done here, elicits spontaneous walking and also activates the neuropils the authors observe.)

We have removed claims of causality from the result section and toned them down in the discussion.

6. The asymmetric activity patterns under turning are a cool result. Here also, I think it would be appropriate to exercise more caution in assigning this to specific neurons.

We have toned down the correspondence to specific neurons

7. Overall, I think the length of some sections in the results diffuses the points the authors want to make. Editing to make it more concise in the results, especially, removing speculation and labeling it more explicitly as speculation in the discussion, would result in a clearer paper.

We have significantly shortened the result section and moved speculative ideas into the discussion. We have also added images of the expression patterns for the different genotypes to give a sense of the distribution of the baseline fluorescence.

Reviewer #2 (Recommendations for the authors):Results:First paragraph: "We expressed GCaMP pan-neuronally (i.e., nsyb-gal4;UAS-GCaMP6s/6m/6f/7s/7f)". The text in the parens is confusing because it does not help the reader understand when different versions of GCaMP were used and why.

We have clarified this point in the text: “We used a variety of GCaMP sensors (UAS-GCaMP6s/6m/6f/7s/7f) and frame rates to capture both high speed and low signal to noise transients.”

Second paragraph: Movie 1-can you comment on the extent to which the presence of the esophagus obscures activity in the subesophageal zone?

We have added a sentence to this effect in the methods section: “Note that the intact esophagus partially obscured the subesophageal zone, decreasing the intensity and resolution in this area. The signal was nevertheless sufficient to identify several subregions with the component analysis (see below).”

Third paragraph: The idea that frame rates between 5-98 Hz are not different for this analysis is surprising. Can you comment on why the frame rate does not seem especially relevant for these experiments? Is this due to the speed of GCaMP6f?

It is possible that there is a difference, but in our current data set and based on the number of experiments we have carried out we did not find a clear indication that this is indeed the case. We have rewritten this sentence to reflect this idea:

“With our data (n=12 flies), we found no significant difference between the global activity R^2^ for walk at different frame rates (5 – 98 Hz) for flies expressing GCaMP6f pan-neuronally”.

It is also possible that the gain in information with speed for a global brain analysis is degraded by a lower signal to noise ratio. Nevertheless, the gain in speed was sufficient to analysis pre-walk activity in certain regions (see below).

Fifth paragraph: Can you expand on the effect of the version of GCaMP?

We have added a short explanation in the text: “We found a small but significant effect of the GCaMP version used, likely reflecting the higher signal to noise ratio for GCaMP6s and GCaMP7s”

Aimon et al. 2019 used norpA mutants to show that global increases in brain activity during walk are independent of visual sensory input. Can you comment on why you repeated this experiment with black nail polish? How do these results enrich our understanding beyond what you have previously published?

The reviewer raises a good point. The black nail polish only confirms with a complementary method what was already shown in Aimon et al., 2019. We have made this clearer in the text:

“As a complementary approach to using norpA mutants in (Aimon et al., 2019), we performed the same experiments but covered the fly’s eyes with black nail polish to prevent outside light from activating its photoreceptor neurons.

Figure S1: Please provide appropriate citations for Figure S1, panel K, and indicate that this image has been reproduced from another publication.

We have added “reproduced from (Ito et al., 2014)” in the legend.

Sixth paragraph: The difference in R2 for global brain activation for different walking substrates seems important but is somewhat glossed over. Can you expand on why you think different walking substrates impact R2 for global brain activation?

The improved model coefficient and R^2^ for the small Styrofoam ball could, we think, indicate that ball movements in this case are a better proxy of the fly movements than the air supported ball which can have erratic movement independent of the fly. Alternatively, the two different surfaces might play a role. We have clarified this in the text:

“The comparison of the two datasets revealed a significant difference in R^2^ for global brain activity (Figure 1-S1H), possibly reflecting differences in surface material, or erratic movements of the air supported ball unrelated to the fly movements confounding the walk regressor.”

Both inhibitory and excitatory neurons are recruited during walk.First paragraph: The way this paragraph is written, the reader is led to believe that you are looking for specific descending neuron types and investigating their neurontransmitter profiles. Consider rephrasing to convey that you are looking at all neuronal classes (interneurons, ascending neurons, descending neurons) and investigating how neurons that release different types of neurotransmitters contribute to global brain activity during walking.

We have removed the part on descending neurons.

Second paragraph: Why change to GCaMP6m here after using GCaMP6f for the global experiments?

The main reason is that the signal-to-noise ratio tends to be lower for more specific lines and requires a more sensitive GCaMP as compared to pan-neuronal expression (see also above).

Figure 2A. Why do you think the R2 during walk for Cha, VGlut, and GAD is higher than it is for pan-neuronal drivers? Is the relatively high R2 for GAD sufficient to conclude that GABAergic neurons explain relatively more of the variance during walk?

As the signal-to-noise for these lines was lower than for the pan-neuronal lines, we used nearly exclusively GCaMP6m (as well as slower frame rates), which likely explains why the R^2^ is overall higher. As the difference between Cha-, Vglut- and Gad-GAL4 was not significant (see legend in Figure 2), we did not comment further on these potential differences.

Aminergic neuron activity is strongly correlated with behaviorFigure 3. To what extent are activity maps for more sparse neuromodulatory neuron types related to the expression patterns of these lines? If you were to activate all neurons in these lines with something like Chrimson (rather than with walking), would you see similar or different subsets of regions activated?

We thank the reviewer for raising this point. In principle, as we are dividing by the basal fluorescence, sparser labeling shouldn’t affect our maps. The resulting difference in signal-to-noise ratio could affect the R^2^ but not the regression coefficients (see also response to reviewer 1). That said it is true that the absence of a certain type of neuron in an area will lead to low R^2^ and coefficients. It is important to note that this is part of the conclusion we are trying to convey with these experiments. In other words, we are showing where neuromodulatory activity could have an impact on wholebrain states. Whether a neuron is absent or not active should not affect this interpretation and the effect on the brain.

To give a better sense to the reader how heterogeneous these expression patterns are, we have added baseline fluorescence images for each genotype in supplement. Furthermore we have added words of caution concerning results from neuropil averages, e.g. in the paragraph on excitatory and inhibitory neurons:

“Future work will be necessary to determine whether such patterns are driven by all neurons expressing a specific neurotransmitter or a subset of neurons in each region.“

Whole brain activity data identifies specific brain regions or even neurons.Fifth paragraph:"Although most component were correlated with walk, other had an R2 indistinguishable from zero thus likely representing ongoing activity unrelated to walk (Schaffer et al., 2021)." Revise to "Although most components were correlated with walk, others had an R2 that was indistinguishable from zero thus likely representing ongoing activity unrelated to walk (Schaffer et al., 2021)."

We have added the “s”.

Turning activates specific brain regions and neurons.First paragraph:"These DNs were previously described by Namiki et al., who generated specific Gal4 lines for neuronal manipulation (Namiki et al., 2018; Robie et al., 2017)." Aren't these split-GAL4 lines?Many neurons have similar morphology, so concluding that activity in IPS-Y and LAL-PS is due to specific DNs should await further experiments. For example, subtracting DNb02 and DNb01 from the GAL4 pattern with LexA expressing GAL80 to show this IPS-Y and LAL-PS activity disappears would be helpful. Alternatively, the authors could image DNb02 and DNb01 split-GAL4 lines to show comparable dynamics of activation during turning."As for forward walk, most components could be matched to candidate neurons (Table 2) for future functional analysis." Generally speaking, it would be helpful to communicate more strongly that any component to candidate neuron matching is tentative. The data presented here is insufficient to conclude that the components can be matched with specific neuronal cell types.

We agree with the reviewer that these propositions (including the one on the DNs above) were very speculative. We have moved most of these statements into the discussion, where we have toned down the conclusions regarding the correspondence between components and neurons.

Brain dynamics at transitions between rest and walk.Figure 6. Adding a y-axis scale (at least at the beginning of each row of graphs) would help to orient the reader. Additionally, more description of this figure in the figure legend is needed, especially to add that the red bar is presumably indicating walk onset?

We have added “Red band indicates walk onset.” In the legend.” Norm dF/F characterizes the y-axis in the beginning of each row.

Increases in activity before walk onset in PRW, posterior slope (IPS-Y, SPS), and GNGm are small and difficult to appreciate. Is there a more rigorous way to quantify the onset of activation, perhaps by looking at the derivative of normalized dF/F?

We agree with the reviewer that more analysis was needed here. To show that some regions are activated before walk onset, we chose to test whether the integral between 500 to 0 ms before walk onset was positive and corrected for multiple comparison. We have added stars indicating significance and details in the legend:

“Stars correspond to above zero significance for integrated traces from -0.5 s to start of walk (Wilcoxon one sided test). Multiple comparison adjusted p-values (Benjamini Hochberg): * p<0.05, ** p<0.01, ***p<0.001.”

"Of note, the component PB-EB corresponds to the previously described head direction cells shown to receive movement-related information during navigation (Lu et al., 2022; Lyu et al., 2022; Seelig and Jayaraman, 2015). EB-DA, found in data from dopaminergic neurons, shows among the strongest and most reliable activity during walk. These neurons were shown to be involved in ethanol-induced locomotion (Kong et al., 2010) and are also involved in sleep regulation (Liang et al., 2016)."How sure are the authors that the PB-EB component corresponds to head direction cells? I would suggest softening this assertion to express some uncertainty unless further confirmation experiments are added.

Indeed, the correspondence between the component and head direction cells is speculative. We have removed this from the result section and softened our claims in the discussion.

"Several other components displayed more variable, and at times too variable dynamics between flies to detect a clear direction (Figure S6)." Which components are the authors referring to? It would be helpful to elaborate on this either in the text or in the Figure S6 legend. For Figure S6, what does the grey vertical bar indicate in each plot?

We have modified the text to “Several other components displayed more variable, and at times too variable dynamics between flies to detect a clear direction (i.e. FBlayv in Figure 6-S1). “. We have also added “Grey bar corresponds to the onset of walk.” into the legend of Figure 6-S1.

"The observed activity patterns confirm and extend published neuronal manipulation data, where available, on sleep and locomotion (see above and discussion)" I would soften this claim to say that these data support published neuronal manipulation data. I do not think these data extend published neuronal manipulation studies as no such manipulations were performed. The data presented here could be used to generate hypotheses for future studies.

We have removed these comments from the Results section and softened them in the discussion.

Forced walk and forced turning recapitulates most activitySecond paragraph: Is the expression of syt-GCAMP stronger in GNG, AMMC, and AVLP (this could be possible if there are more synapses in these regions)? If you activated the brain pan-neuronally with Chrimson and imaged using syt-GCaMP, would you see a different pattern of activity than what is seen suiting walk?

This experiment is interesting but has a strong confound that levels of neuronal activation with Chrimson are unlikely to match natural activation. Instead, we compared the basal fluorescence for syt-GCaMP with the activity pattern (see image of *nsyb*-GAL4, UAS-nsyt-GcaMP6s expression pattern in Figure 7-S1) as lower expression could lead to a lower signal-to-noise ratio and thus a lower R^2^ value. Note that the effect on the correlation coefficient (that we used in complement with R^2^ in Figure 7-S1) shouldn’t be affected by this lower signal-to-noise ratio in the same way. Nevertheless, we added the expression pattern in supplementary Figure 7-S1 as it can help interpreting the data.

For Figure 7G/ the difference between spontaneous and forced walk, it would be helpful to add statistical comparisons of the normalized activity traces when claiming that they are different versus the same.

We have performed this analysis.

Discussion:Role and origin of broad activation during ongoing behaviorFirst paragraph: Delete the underscore between "coupling_sensory"

We have removed the underscore.

Role and origin of broad activation during ongoing behaviorFifth paragraph: "More generally, our data indicate that regions triggering walk are also highly sensitive to walk itself suggesting a closed-loop between walk initiation and walk maintenance, possibly including factors such as speed, detailed movement etc." The data presented here is insufficient to conclude that these regions are responsible for triggering walk without further neuronal manipulation studies. The authors should support this claim by showing that MB components correspond to specific neuronal cell types (by removing specific neurons from the global pattern or imaging specific neurons and showing that they show identical dynamics) and then activate and silence these specifical neuronal cell types while analyzing walking behavior.

We have toned down claims of causality throughout the manuscript. For example: “This suggests a potential role of these areas in initiating walk. Activity before walk onset could also be due to preparatory movements that were not detected as walk (Ache et al., 2019), or represent the activity of neurons downstream of neurons responsible for triggering walk.”

Reviewer #3 (Recommendations for the authors):1) The word "underpin" in the title may be too strong because it can be construed to imply a degree of causality, which is not shown. I suggest more precise phrasing like "correlates with" or something similar that clearly does not claim causal relations.

What we meant here is that whole brain state change is composed of changes in activity in all brain regions and neuron types. We changed the title to “Global change in brain state during spontaneous and forced walk in *Drosophila* is composed of combined activity patterns of different neuron classes”.

2) Throughout the text activity correlations of many dozens of subregions are compared. E.g. Figure 3F tests 40 hypotheses. Figure 5b tests closer to 90. With so many brain regions it will be important to leverage a statistical framework that explicitly considers multiple hypothesis testing in order to better convey to a reader the relative confidence in each finding. Some avenues to consider: a q-value analysis could be used that articulates a false-discovery rate. Shuffled datasets can also be used to try to reject null hypotheses. And GFP-only animals that lack GCaMP could also be used to generate null distributions to compare against and to reject null hypotheses.

We have corrected the figures to reflect significance after multiple comparison correction.

3) Figure S7: the gray shading and blue traces are not defined. At first, I had assumed gray is an error bar and blue is average, but if that's the case, there appears to be a major problem in Figure S7C where these diverge dramatically, e.g. in SMB and SPS and NO.

We thank the reviewer for noticing this error in the plots in Figure 6-S2. We have corrected it and added a description of the in the legend: “Grey shading corresponds to the standard deviation. “